# The Data Manifold under the Microscope

**Marios Koulakis** [1]   **Constantin Seibold** [2]

## Abstract

A significant gap exists between theory and practice in deep learning. Generalization and approximation error bounds are often derived for simplified models or are too loose to be informative. Many rely on the manifold hypothesis and on geometric regularity such as intrinsic dimension, curvature, and reach. Progress requires insight into data-manifold geometry and suitable benchmarks, yet existing options are polarized: analytic manifolds with known geometry but limited applicability, or real-world datasets where geometry is only coarsely estimable. We introduce a benchmarking framework for studying data geometry. We repurpose and extend dSprites and COIL-20 with additional transformation dimensions and dense, axis-aligned sampling, and pair them with finite-difference estimators that recover curvature, reach, and volume at near-ground-truth accuracy in a regime where general-purpose estimators are unreliable or difficult to deploy. The framework is intended as a controlled testbed, useful as a calibration environment for geometric estimators and a sandbox for probing theoretical assumptions. To illustrate its use, we present two application studies, namely assessing the scaling behavior of the bounds of Genovese et al. and Fefferman et al., and tracking the layer-wise geometry of a $\beta$-VAE, highlighting the behavior of current bounds and the value of controlled benchmarks for guiding and validating future theory.

A reference implementation is available at github.com/koulakis/manifold-microscope.

[1]Independent Researcher, Germany [2]Diagnostic and Interventional Radiology, University Clinic Heidelberg, Heidelberg, Germany. Correspondence to: Marios Koulakis <mariosco100@gmail.com>.

*Proceedings of the 43$^{rd}$ International Conference on Machine Learning*, Seoul, South Korea. PMLR 306, 2026. Copyright 2026 by the author(s).

## 1. Introduction

Deep learning has become the dominant paradigm for a broad range of tasks, and in recent years generative models in particular, such as variational autoencoders (VAEs) (Kingma & Welling, 2013), diffusion models (Ho et al., 2020), and modern masked or autoencoding architectures (He et al., 2022), have seen striking empirical success. A common way to interpret this success is through the manifold hypothesis (Cayton et al., 2005): high-dimensional data often concentrate near low-dimensional manifolds, and learning amounts to finding useful parameterizations of them. Many modern models can thus be viewed as procedures for fitting or approximating data manifolds, whether explicitly, as in latent-variable models (Arvanitidis et al., 2017), or implicitly, as in denoising and score-based flows (Horvat & Pfister, 2021).

While appealing, it remains unclear how networks fit manifolds in practice. Classical results such as (Genovese et al., 2012) and (Fefferman et al., 2018) on minimax rates for manifold estimation, (Aamari & Levrard, 2019) on the tightness of those bounds under varying manifold smoothness, and more recent work on approximation of Sobolev classes on manifolds (Tan et al., 2024) highlight the role of geometric quantities like curvature, reach, or sampling density in learning. Yet in realistic data these quantities are rarely observable: the data-generating process is unknown, the intrinsic dimension is uncertain (Campadelli et al., 2015), and sampling can be irregular (Shroff et al., 2011; Sedghi et al., 2020). As a result, theoretical guarantees often rely on constants that cannot be directly checked or measured.

This creates two complementary needs. On the theoretical side, sharper and more data-adaptive bounds are desirable, or at least a clearer picture of when existing bounds are informative. On the empirical side, there is a lack of benchmarks that balance realism with geometric control: synthetic manifolds are too simple, while real-world datasets obscure the geometry entirely.

To narrow this gap, we introduce a framework that constructs low-dimensional image families densely sampled along controlled transformation axes (e.g., rotation, translation, scale) and provides efficient finite-difference estimators of geometric measures such as curvature, reach, and volume. Combined with an experimental pipeline for man-

ifold fitting, it enables systematic tests of how theory and practice align. While we illustrate it here using generative models and manifold-fitting bounds, the framework is general and also supports settings such as discriminative learning or benchmarking geometric-measure estimators. Our contributions are the following:

- A reproducible framework of adapted low-dimensional datasets with dense, axis-aligned sampling, plus an experimental pipeline for probing modern generative and representation models.

- A suite of efficient finite-difference estimators for pointwise and global geometric quantities such as curvature, reach, or volume of manifolds.

- Two illustrative case studies that show possible uses of the framework, empirically probing how the scaling of existing manifold-fitting bounds matches observed errors, and tracking how a $\beta$-VAE reshapes manifold geometry across layers.

## 2. Related Work

**Manifold Analysis:** Following the manifold hypothesis (Cayton et al., 2005), many works analyze the geometry of data and learned representations. Early results on inferring topological structure from samples include guarantees for homology recovery (Niyogi et al., 2008). Studies on VAEs and $\beta$-VAEs (Kingma et al., 2019; Higgins et al., 2017) show that learned latent spaces often exhibit limited curvature (Arvanitidis et al., 2017; Shao et al., 2018). Geometric invariants such as curvature, tangent spaces, and reach have been used to study robustness and disentanglement (Aamari et al., 2019; Berenfeld et al., 2022; Birdal et al., 2021), though outcomes depend strongly on dataset, architecture, and estimator (Brahma et al., 2015; Kaufman & Azencot, 2023). Recent representation-learning approaches explicitly impose manifold structure via learned charts (Schonsheck et al., 2019). Connections to expressive power have also been explored through manifold topology (Yao et al., 2024a).

**Manifold Fitting Bounds:** (Genovese et al., 2012) established the minimax rate $O((1/n)^{2/(2+d)})$, later shown optimal by (Kim & Zhou, 2015), though the corresponding estimator is computationally infeasible. (Aamari & Levrard, 2019) provides nonasymptotic rates for manifold estimation that depend on manifold smoothness, alongside rates for tangent space and curvature estimation. (Yao & Xia, 2025) handled unbounded noise via projection, yielding $O(\sigma^2 \log(1/\sigma))$ Hausdorff error for sample size $O((1/\sigma)^{d+3})$. Neural estimators (Yao et al., 2023; 2024b) offer scalable alternatives at the cost of weaker guarantees. Earlier complexity results for testing the manifold hypothe-

sis (Narayanan & Mitter, 2010) relate geometry, dimension, and sample efficiency more explicitly.

A complementary line of work uses reach as a structural primitive. (Fefferman et al., 2016) introduced a geometric approach based on preserving reach, with noisy-data extensions reducing complexity from double- to single-exponential (Fefferman et al., 2018; 2020). However, these guarantees depend on unknown geometric parameters such as reach and volume. For example, the constants in (Yao & Xia, 2025) scale as $\tau^{-2}$, illustrating a recurring issue: theoretical bounds often require prior knowledge of quantities that can only be estimated from the manifold itself.

**Geometric Property Estimation:** Recent work develops estimators and sample complexity bounds for reach and related quantities (Aamari et al., 2019; Berenfeld et al., 2022; Aamari et al., 2023), though empirical validation remains challenging due to the lack of ground truth geometric data. Similar progress exists for curvature estimation: scalar curvature estimators with nonasymptotic guarantees (Aamari & Levrard, 2019; Gawlik & Neunteufel, 2025), methods for the second fundamental form and Ricci curvature (Acosta et al., 2023; Samal et al., 2018), and tangent space estimators with provable accuracy (Cheng & Chiu, 2016; Cazals & Pouget, 2005). These results extend the classical geometric recovery literature (Niyogi et al., 2008).

Overall, theory provides strong asymptotic guarantees, but empirical understanding is hindered by the absence of datasets with exact geometric ground truth. Our framework addresses this gap by combining dense, axis-aligned sampling with finite-difference estimators, providing a controlled calibration testbed in which geometric estimators can be validated and theoretical predictions can be probed at low intrinsic dimension.

## 3. Background and Definitions

**Datasets as Manifolds:** The manifold hypothesis suggests that high-dimensional datasets often concentrate near low-dimensional manifolds. We focus on datasets where the number of intrinsic factors of variation $d$ is small and fixed, and where these factors are explicitly known. Each dataset is modeled as a union of connected smooth $d$-dimensional manifolds embedded in $\mathbb{R}^D$, possibly with boundary. Typically, each of the $k$ semantic classes in the dataset corresponds to a separate connected component of this union.

For this work, we restrict attention to simple topologies in which each manifold factors into cyclic and non-cyclic directions. Concretely, every manifold is assumed to be homeomorphic to $[0, 1]^r \times (S^1)^s, r + s = d$, so that some coordinates vary over a compact interval while others wrap around a circle. This setting captures many common synthetic datasets. A canonical example is *dSprites*, a collection

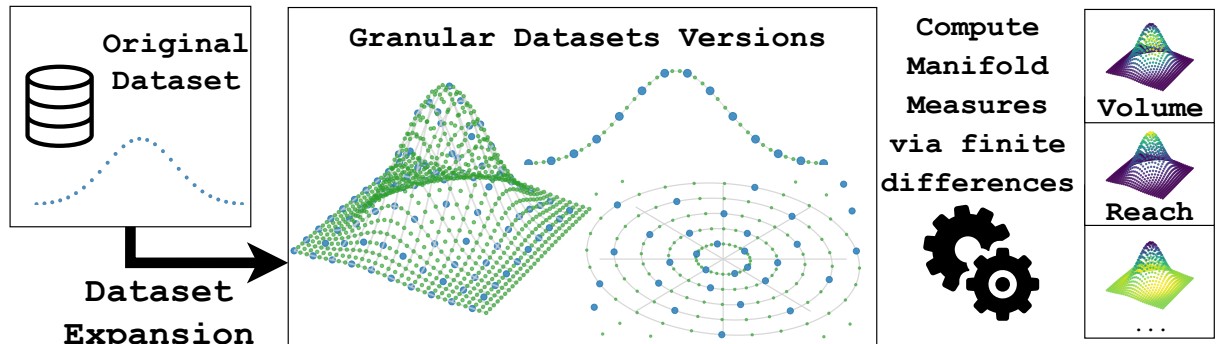

*Figure 1.* From a low-dimensional seed we produce dense, regular grids via analytic parametrizations or systematic transforms; central finite differences recover geometric estimators such as reach and curvature enabling validation of manifold-fitting bounds.

of $64 \times 64$ grayscale images of objects (square, ellipse, heart) undergoing controlled transformations such as scaling, rotation, and translation. The manifold of a class in this dataset is homeomorphic to $[0,1]^3 \times S^1$ and is embedded into $\mathbb{R}^{4096}$. Note that manifolds of any topology can also be handled by covering them with charts and applying the framework chart-wise.

To obtain discrete datasets from this continuous geometric picture, we impose a grid structure. Let $n_l$ be the number of sampled values along the $l$-th dimension, so that the total number of grid points is $n = \prod_{l \leq d} n_l$. We define

$$G = \left\{ \left( \tfrac{j_1}{n_1 - 1}, \dots, \tfrac{j_r}{n_r - 1}, 2\pi \tfrac{j_{r+1}}{n_{r+1}}, \dots, 2\pi \tfrac{j_d}{n_d} \right) \Big| j_\ell \in [0, n_\ell) \right\} \tag{1}$$

where the first $r$ coordinates and the last $s$ coordinates sample $[0,1]$ and $S^1$ respectively on constant intervals. For each class $i \leq k$, we define a mapping $u_i : G \to M_i$ that associates each grid point with a dataset element obtained by applying the corresponding transformations. The image $u_i[G]$ provides a discrete parametrization of the manifold $M_i$, and by cutting along cyclic directions one obtains discrete patches of $M_i$. The complete discretized dataset is then

$$X_G = \bigcup_{i \leq k} u_i[G]. \tag{2}$$

**Geometric Measures:** We focus on three geometric quantities describing the local and global structure of a manifold: *volume*, *scalar curvature*, and *reach*. For completeness, we recall the definitions of the Riemannian metric and the objects needed to introduce these quantities.

**Definition 3.1** (Riemannian metric). A Riemannian metric $g$ on a $d$-dimensional differentiable manifold $M$ assigns to each point $p \in M$ an inner product $g_p$ on the tangent space $T_p M$. If $M$ is embedded in $\mathbb{R}^D$, the ambient Euclidean metric induces a Riemannian metric by restriction. In local coordinates $u : U \to M$, the metric matrix is

$$g_{ij}(x) = \langle \partial_i u(x), \partial_j u(x) \rangle_{\mathbb{R}^D}. \tag{3}$$

**Definition 3.2** (Volume element and volume). On a Riemannian manifold $(M, g)$ with local coordinates $(u^1, \dots, u^d)$, the natural volume element is

$$d\,V = \sqrt{\det(g)}\, du^1 \wedge \cdots \wedge du^d. \tag{4}$$

The volume of a region $R \subset M$ is then

$$\mathrm{Vol}(R) = \int_{u^{-1}(R)} \sqrt{\det(g)}\, du^1 \cdots du^d. \tag{5}$$

Throughout, we use Einstein summation notation: free upper and lower indices in the same term are implicitly summed, e.g. $g^{ij} Ric_{ij} = \sum_{i,j} g^{ij} Ric_{ij}$.

**Definition 3.3** (Notions of curvature). The curvature of $(M, g)$ is encoded in the Riemann tensor

$$R^i_{\ jkl} = \partial_k \Gamma^i_{jl} - \partial_l \Gamma^i_{jk} + \Gamma^i_{kr} \Gamma^r_{jl} - \Gamma^i_{lr} \Gamma^r_{jk}, \tag{6}$$

where the Christoffel symbols are

$$\Gamma^i_{jk} = \tfrac{1}{2} g^{ir} (\partial_j g_{rk} + \partial_k g_{rj} - \partial_r g_{jk}). \tag{7}$$

Contracting yields the Ricci tensor $Ric_{ij} = R^r_{\ irj}$ and the scalar curvature $R = g^{ij} Ric_{ij}$.

For more details on the above definitions one can refer to a classical differential geometry reference, for example (Do Carmo, 1992).

**Definition 3.4** (Reach (Federer, 1959)). For a closed set $A \subset \mathbb{R}^D$, the *medial axis* is the set of points with more than one nearest neighbor in $A$. The *reach* of $A$ is the distance to its medial axis:

$$\tau_A = \inf_{p \in A} d_{\ell^2}(p, Med(A)). \tag{8}$$

Here $d_{\ell^2}$ is the Euclidean distance: $d_{\ell^2}(p, q) = \|p - q\|_2$, and for a set $A$, $d_{\ell^2}(p, A) = \inf_{q \in A} d_{\ell^2}(p, q)$. Intuitively, $\tau_A$ is the largest radius such that every point within distance $< \tau_A$ of $A$ has a unique nearest neighbor in $A$. For a

compact $d$-dimensional submanifold $M \subset \mathbb{R}^D$, the reach can be expressed as

$$\tau_M = \inf_{p \neq q \in M} \frac{\|p - q\|^2}{2\, d_{\ell^2}(q - p, T_p M)}, \qquad (9)$$

where $d_{\ell^2}(v, T_p M)$ is the distance of a vector $v$ to the tangent space at $p$.

In our applications, we consider both *global* quantities (total volume, the scalar curvature integral[1], global reach) and their *local* counterparts (volume element, pointwise scalar curvature, and local reach estimators as in (Aamari et al., 2019)).

**Definition 3.5** (Hausdorff distance). Let $A, B \subset \mathbb{R}^D$ be non-empty subsets. The *Hausdorff distance* between $A$ and $B$ is

$$H(A, B) = \max \left\{ \sup_{a \in A} d_{\ell^2}(a, B), \sup_{b \in B} d_{\ell^2}(b, A) \right\}. \quad (10)$$

If $M$ is the ground-truth data manifold and $\hat{M}$ a learned approximation (e.g., via an autoencoder), then $H(M, \hat{M})$ expresses the largest geometric error, i.e. how far the manifolds can be from each other at any single point.

**Manifold Fitting and Bounds:**

Manifold learning and generative modeling are closely related: both assume high-dimensional data lie near a low-dimensional manifold. Generative models learn a mapping between the data manifold in $\mathbb{R}^D$ and a lower-dimensional latent space where factors of variation are (ideally) disentangled and sampling is easier. If the encoder–decoder is sufficiently regular, composing the encoder with the manifold charts $u_i$ gives a parametrization of the latent manifold. Data points map to latent representations, and decoding maps them back to the estimated data manifold.

We focus on two theoretical results that provide bounds on the error of manifold fitting in terms of sample size, intrinsic dimension, and geometric properties of the manifold. Before stating these results, we define the normal fiber of radius $r$ at a point $p \in M$ as $L_r(p) = (p + T_p^\perp M) \cap B_D(p, r)$, where $T_p^\perp M$ is the orthogonal complement of $T_p M$ (viewed as a linear subspace of $\mathbb{R}^D$) and $B_D(p, r)$ is the $D$-dimensional ball of radius $r$ centered at $p$. Intuitively, this fiber is a $(D - d)$-dimensional ball extending away from the manifold at $p$.

**Minimax manifold estimation bound:** (Genovese et al., 2012) establish minimax rates for estimating a manifold from samples corrupted by small normal noise. Their result shows that the intrinsic dimension $d$ alone governs the difficulty of estimation.

---

[1]The integral of the scalar field defined by the scalar curvature $R$ on $M$ over the volume of $M$, i.e. $\int_M R\, dV$.

**Theorem 3.6** ((Genovese et al., 2012)). *Let $\mathcal{M}(\tau)$ be the class of compact, smooth, boundaryless $d$-dimensional manifolds embedded in $\mathbb{R}^D$ with reach at least $\tau$. Fix $0 < \sigma < \tau$. For each $M \in \mathcal{M}(\tau)$, let $Q_M$ be the distribution of $Y = \xi + Z$, where $\xi$ is uniformly distributed on $M$ and, conditional on $\xi$, $Z$ is uniformly distributed on $\{z \in T_\xi^\perp M : \|z\|_2 \leq \sigma\}$. Let $\mathcal{Q} = \{Q_M \mid M \in \mathcal{M}(\tau)\}$. For an $n$-sample estimator $\widehat{M} : (\mathbb{R}^D)^n \to \mathcal{M}$, define the minimax risk*

$$R_n(\mathcal{Q}) = \inf_{\widehat{M}} \sup_{Q \in \mathcal{Q}} \mathbb{E}_Q[H(\widehat{M}, M)], \qquad (11)$$

*where $H(\cdot, \cdot)$ denotes the Hausdorff distance. Then there exist constants $C_1, C_2 > 0$ such that*

$$C_1 \left(\frac{1}{n}\right)^{\frac{2}{2+d}} \leq R_n(\mathcal{Q}) \leq C_2 \left(\frac{\log n}{n}\right)^{\frac{2}{2+d}}. \qquad (12)$$

This result shows that the sample complexity depends exponentially on the intrinsic dimension $d$, but not on the ambient dimension $D$. Intuitively, the distributions $Q_M$ correspond to sampling a point on $M$ and perturbing it orthogonally within its reach. In our setting, an estimator $\widehat{M}$ may be interpreted as a neural network (e.g., an autoencoder) that outputs a fitted manifold.

**Bounds from Fitting a putative manifold to noisy data:** (Fefferman et al., 2018) give an upper bound for the Hausdorff distance between the true manifold $M$ and an estimated manifold $M_o$ produced by their algorithm. Specializing their bound to the low-noise regime (or the noiseless case) yields the scaling

$$H(M_o, M) < C_1 \left(\frac{\log n}{n}\right)^{\frac{1}{d}}, \qquad (13)$$

where $C_1$ depends on geometric parameters of the class (e.g. volume and sampling density; see Appendix A for details).

**The exponents of the bounds:** In the noise-free (or small-noise) regime, (Aamari & Levrard, 2019) provide rates for manifold estimation under Hausdorff loss that depend on the smoothness order $k$. In particular, for manifolds in $\mathcal{C}^k$ (with $k \geq 3$), their results yield the scaling

$$R_n(\mathcal{Q}) < C_1 \left(\frac{\log n}{n}\right)^{\frac{k}{d}}. \qquad (14)$$

Beyond improving rates, their results suggest that the exponent is tied to the order of the local fit: roughly speaking, a linear, e.g. PCA-based, method corresponds to an exponent $1/d$, while a quadratic local fit corresponds to an exponent $2/d$.

In our experiments, we compare the dimension-dependent minimax scaling of (Genovese et al., 2012) with the linear-fit scaling suggested by (Fefferman et al., 2018).

*Table 1.* Dataset characteristics and parametrizations used in our experiments:

| Dataset | $d$ | $D$ | Factors | Range | Components |
|---|---|---|---|---|---|
| $S^1$ | 1 | 2 | $\phi_1$ | $[0, 2\pi)$ | 1 |
| Two moons | 1 | 2 | $\phi_1$ | $[0, \pi)$ | 2 |
| $S^2$ | 2 | 3 | $\phi_1, \phi_2$ | $[0, \pi) \times [0, 2\pi)$ | 1 |
| $T^2$ | 2 | 3 | $\phi_1, \phi_2$ | $[0, 2\pi) \times [0, 2\pi)$ | 1 |
| dSprites | 4 | 4096 | scale, orientation, pos. x, pos. y | see Appendix E | 3 |
| COIL-20 | 3 | 4096 | horiz. orient., scale, img. orient. | see Appendix E | 20 |

## 4. Methods and Framework

We construct controlled synthetic manifolds of low intrinsic dimension ($d = 1$–$4$) sampled on regularly spaced grids. Dense sampling enables stable finite-difference approximations of partial derivatives, allowing accurate computation of the induced metric, volume element, curvature tensors, and reach. This framework provides a setting to empirically assess theoretical manifold-fitting bounds and a reproducible dataset suite for validating geometric estimators.

**Datasets with a dense grid structure:** We consider two complementary approaches for constructing low-dimensional datasets with a grid structure.

For analytically defined manifolds, we generate a regular grid using known parametrizations. For image-based or domain-specific datasets, we create an analogous structured grid by systematically applying transformations (translations, rotations, scalings) and sampling all combinations.

Grid sampling is dense but not necessarily uniform in intrinsic geometry. To obtain more uniform subsets, we use:

- Iterative farthest-point sampling on the grid to select well-separated points.

- Sampling according to the volume form, either by reweighting grid points or, when analytic expressions are available, via inverse transform sampling using numerical integration, which becomes costly as $d$ grows.

For experiments, we fix a uniformly distributed test subset of grid points and use the remainder for training with varying sizes. This yields clear train/test separation while keeping the test set representative for approximating Hausdorff and average distances. Geometric measures are computed on the full dataset.

**Computation of geometric measures:** To estimate geometric quantities such as the volume element, scalar curvature, and reach, we use finite-difference approximations of partial derivatives on the dense grid. We assume $\mathcal{C}^3$ smoothness for estimating the reach and volume and $\mathcal{C}^5$ smoothness for estimating scalar curvature, enabling second-order finite-difference approximations of the required first- and third-order derivatives.

Given a parametrization $u : \mathbb{R}^d \to \mathbb{R}^D$, the central difference approximation

$$f'(x) = \frac{f(x+h) - f(x-h)}{2h} + O(h^2) \qquad (15)$$

extends coordinate-wise. For example,

$$\frac{u(x_1, \ldots, x_i + h, \ldots, x_d) - u(x_1, \ldots, x_i - h, \ldots, x_d)}{2h}, \qquad (16)$$

gives a second-order accurate estimate of the $i$-th partial derivative $D^c_{h,i} u(x)$. Products of such approximations yield $g_{ij} = \langle u_{,i}, u_{,j} \rangle + O(h^2)$ and the entries of $g^{-1}$ inherit the same order provided the metric is uniformly non-degenerate, e.g. $\lambda_{\min}(g) \geq c > 0$ on the chart. Furthermore, by the $\mathcal{C}^5$ smoothness, all partial derivatives of order up to 3 also have $O(h^2)$ error, therefore all the volume form, Christoffel symbols, curvature tensors and scalar curvature have $O(h^2)$ error. These intermediate tensors are available within the framework for downstream analysis. Concretely, the pointwise volume element and scalar curvature satisfy $|\hat{v} - v| = O(h^2)$ and $|\hat{R} - R| = O(h^2)$ and the corresponding global volume estimate inherits this rate under the quadrature scheme used here. For reach, we use the plug-in estimator of (Aamari et al., 2019) with tangent spaces $T_x$ estimated from the grid by finite differences. Unlike volume and curvature, reach is a minimum over point pairs, so its convergence is not controlled only by the finite-difference accuracy of the local derivatives. In the setting with known tangent spaces, Aamari et al. distinguish a global-bottleneck regime with rate $O(n^{-1/d})$ from a slower local-curvature regime with upper rate $O(n^{-2/(3d-1)})$ for inverse reach, where $n$ is the total number of sample points. Since the reach is bounded away from zero in our setting, inverse-reach and reach errors are equivalent up to constants.

For more details on the derivation of the approximations of the curvature-related measures, please look at App. B.

**Comparison to general estimation methods:**

General estimators of curvature and reach from point clouds, such as those of (Aamari et al., 2023) for reach and (Aamari & Levrard, 2019) for curvature, operate in far broader

settings than ours. They must infer derivatives from unstructured samples via sophisticated interpolations, leading to more complex rates, nontrivial constants, and substantial computational overhead. Unfortunately, few implementations are currently available.

On a quasi-uniform $d$-dimensional grid with $n$ total points, the per-axis grid spacing satisfies $h \asymp n^{-1/d}$. Thus the global-bottleneck and local-curvature rates correspond respectively to

$$O(n^{-1/d}) = O(h) \text{ and } O(n^{-2/(3d-1)}) = O(h^{2d/(3d-1)}).$$

Our finite-difference tangent estimates have error $O(h^2)$. Using the tangent-perturbation stability bound of (Aamari et al., 2019), this contributes a term of order $O(h)$ on a grid with minimum separation $\delta \asymp h$. Hence, in the conservative local-curvature regime, the dominant term remains

$$|\hat{\tau} - \tau| = O(n^{-2/(3d-1)}) = O(h^{2d/(3d-1)}),$$

up to constants and logarithmic factors.

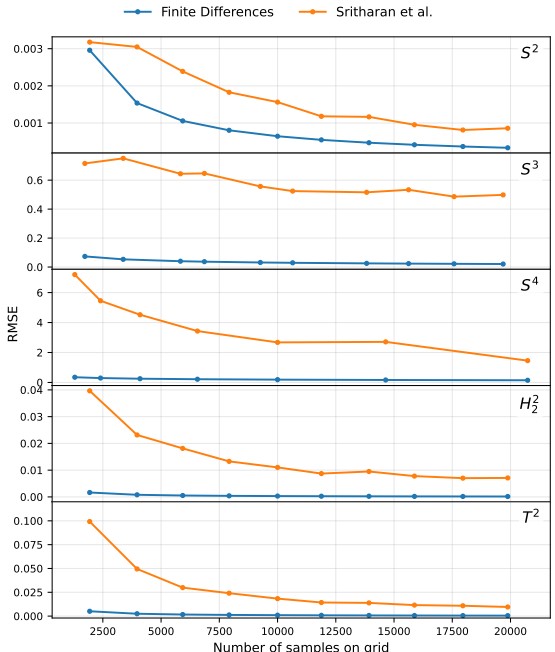

*Figure 2.* Scalar-curvature RMSE (lower is better) versus number of grid samples, comparing our finite-difference estimator (blue) with (Sritharan et al., 2021) (orange) on hyperspheres of multiple dimensions $S^2, S^3, S^4$, the hyperboloid $H_2^2$, and the torus $T^2$.

For scalar curvature, our intrinsic finite-difference estimator gives $|\hat{R} - R| = O(h^2) = O((1/n)^{2/d})$. The theoretical rate of (Aamari & Levrard, 2019) is $O((\log n/n)^{3/d})$ for $C^5$ manifolds, which is slightly tighter. This gap is due to our intrinsic computation of the Riemann curvature tensor, which requires third-order derivatives. An extrinsic formulation via the second fundamental form would reduce our

smoothness requirement to $C^4$ and thus improve the corresponding $O((\log n/n)^{2/d})$ theoretical rate of (Aamari & Levrard, 2019). For details on the derivation of the sample complexities, see appendix B.

As a sanity check, we compare the finite-difference approach with the scalar-curvature estimation method introduced in (Sritharan et al., 2021) on the manifolds of known curvature benchmarked in their paper. For each manifold and for different numbers of samples, we run both approaches and compare their RMSE against the known curvature; see Fig. 2. For the method of (Sritharan et al., 2021), we additionally performed a grid search over the sensitive radius hyperparameter and (optimistically) report the per-point oracle choice, i.e. the radius minimizing the error at each point. Nevertheless, the finite-difference method, aware of the grid structure, yields substantially more accurate estimates.

Since such ground-truth values are not available for the image-based datasets, we separately study their sensitivity to grid resolution in Appendix D using convergence across consecutive grid refinements.

Please note again, our aim is not to compete with general estimators. Instead, the controlled grid setting provides a straightforward and reproducible way to compute geometric quantities at near-optimal accuracy. The resulting measures serve as reliable benchmarks and unit tests for developing and analyzing more general methods on unstructured data.

## 5. Applications

**Datasets:** Our experimental setup employs two types of datasets: (i) toy manifolds with explicit mathematical parametrizations for analytical validation, and (ii) adapted versions of established image datasets (dSprites (Matthey et al., 2017) and COIL-20 (Nene et al., 1996)) that provide controlled settings for empirical evaluation. Table 1 summarizes the key characteristics of each dataset.

The toy manifolds provide analytically tractable benchmarks with closed-form geometric properties. For the image datasets, we adapted the sampling to better support geometric computations: we regenerated dSprites at the same resolution with improved anti-aliasing, and extended COIL-20 with additional transformations such as scale and orientation. Both datasets use dense grid sampling with margin oversampling on non-cyclic dimensions to support finite-difference estimates of geometric measures. Details are in Appendix E.

**Manifold fitting methods:** We consider two complementary approaches to fit manifolds to the datasets: a geometric method, *Manifold Moving Least Squares* (MMLS) (Sober & Levin, 2020), and a deep learning method, the $\beta$-VAE autoencoder (Higgins et al., 2017).

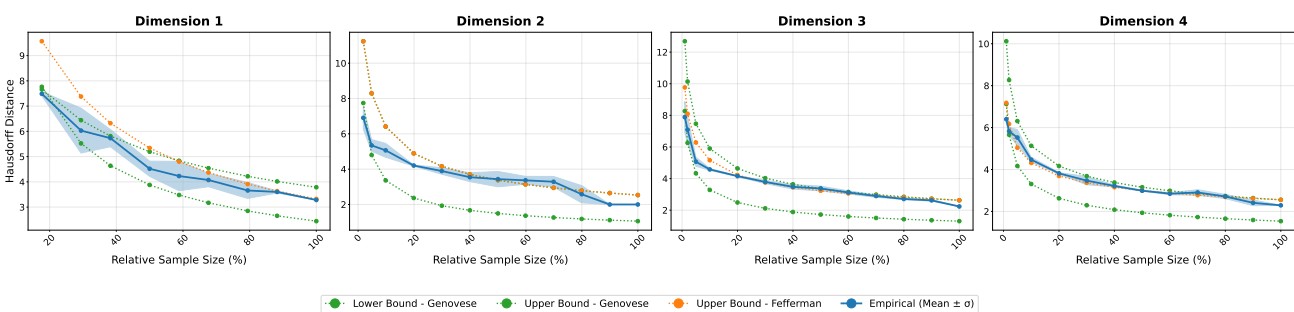

*Figure 3.* Empirical Hausdorff error against theoretical bounds on dSprites for d = 1 to 4, using MMLS (linear local fit). Empirical mean ± $\sigma$ over 3 repetitions in blue, x-axis is training-set fraction (%). Genovese lower bound in dotted green, Genovese upper in solid green, Fefferman upper in orange. Constants fitted to envelope the empirical curve. At d = 2 the Genovese upper and Fefferman bounds overlap.

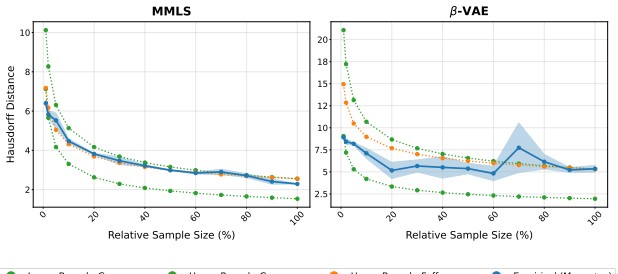

*Figure 4.* Hausdorff error and bounds on 4D dSprites for MMLS (left) and $\beta$-VAE (right; model B, latent dim 10, $\beta$ = 4). Empirical mean ± $\sigma$ over 3 repetitions. Bounds as in Figure 3.

*MMLS:* MMLS (Sober & Levin, 2020) is a local manifold approximation method in which, for each query point, a neighborhood is weighted by a kernel function and a $d$-dimensional affine space together with a local polynomial are fitted by weighted least squares. The projection onto the approximated manifold is then obtained by projecting onto the fitted $d$-plane and evaluating the polynomial correction, yielding a flexible higher-order local model of the manifold.

In our setting, the ambient dimension is too high to reliably estimate local polynomials. We therefore customize the procedure by restricting the fit to the weighted $d$-plane alone. More details on the method and our usage are in Appendix G.1.

$\beta$-*VAE:* The $\beta$-VAE (Higgins et al., 2017) provides a learning-based approach. It maps input data into a low-dimensional latent space and reconstructs them, with the reconstruction lying on a learned manifold. The reconstruction distance can thus be interpreted as the distance to this manifold. As the input dimensionality is too small for meaningful bottlenecks in toy datasets, we employ $\beta$-VAE only for image datasets (dSprites, COIL-20). We use the off-the-shelf hyperparameter configuration of (Higgins et al., 2017). For more details, refer to Appendix G.1.

**Sampling protocol:** For both methods, a fixed test set of 500 uniformly spread points is used. On the toy datasets, training sets range from 5–500 points, and each experiment is repeated 20 times. On dSprites and COIL-20, data ratios range from 0.01 up to 1.0, with three repetitions per setup.

The motivation for considering both methods is that MMLS directly fits a manifold in the original data space, whereas $\beta$-VAE learns a latent manifold with richer semantic structure. Together, they provide complementary views of manifold fitting performance.

**Error bounds evaluation:** We now turn to the first application of manifold fitting: the empirical evaluation of theoretical bounds on approximation error. Recall from Section 3 that we consider the upper and lower bounds from (Genovese et al., 2012) and the upper bound from (Feffer-

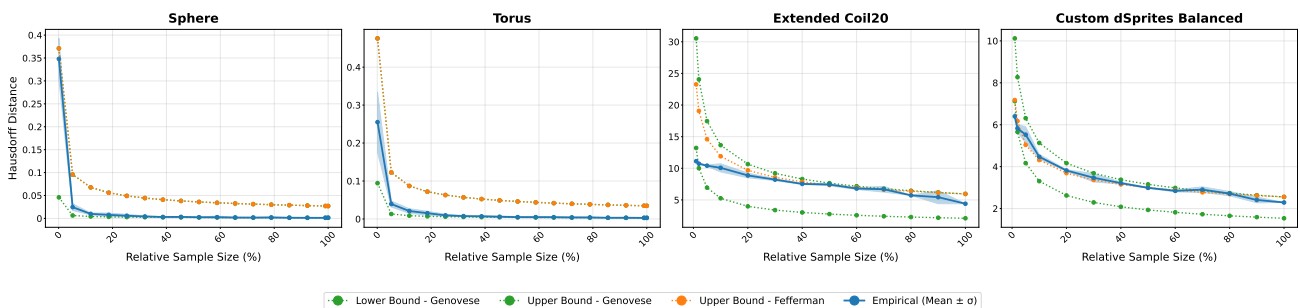

*Figure 5.* Hausdorff error and bounds for MMLS on, from left to right, Sphere $S^2$, Torus $T^2$, extended COIL-20, and balanced dSprites. Empirical mean ± $\sigma$ over 3 repetitions, x-axis is training-set fraction (%). Please note that for $S^2$ and $T^2$, the Genovese upper and Fefferman bounds are the same for $d = 2$ since we use the same fitting process for their respective multiplicative constants.

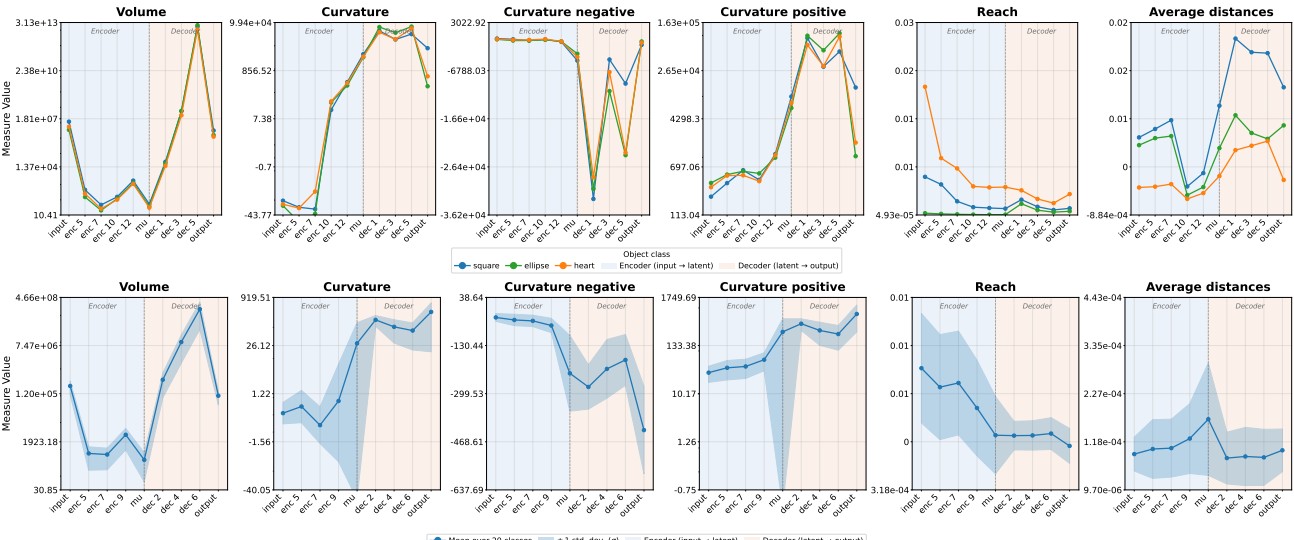

*Figure 6.* **Evolution of manifold geometry across the layers of a $\beta$-VAE.** For each layer (input $\rightarrow$ encoder $\rightarrow$ latent $\mu$ $\rightarrow$ decoder $\rightarrow$ output; encoder and decoder halves shaded blue and orange) we report six geometric measures: volume, scalar curvature, its negative and positive parts, reach, and average pairwise distance (all normalized except volume). *Top:* dSprites, one curve per object class (square, ellipse, heart). *Bottom:* COIL-20, summarized as the mean over the 20 object classes (solid line) with a $\pm 1\sigma$ band (shaded: one standard deviation across classes). The full per-class COIL-20 version is given in Fig. 31.

man et al., 2018). For the fitting methods (MMLS, $\beta$-VAE), each dataset (toy, dSprites, COIL-20), and a range of intrinsic dimensions, we fit manifolds for multiple training set sizes. As an evaluation metric we use the directed Hausdorff distance from the ground-truth manifold to the fitted manifold, $\sup_{x \in M} d_{\ell^2}(x, \widehat{M})$, which provides a computable proxy for $H(M, \widehat{M})$; the reverse direction is discussed in Appendix G.1.

**Comparing to bounds:** To estimate the constants appearing in the theoretical upper and lower error bounds, we avoid attempting to compute these constants directly. Instead, we determine them empirically by aligning the theoretical shapes of the bounds with the observed error curve. For a full justification, see Appendix G.2.

The theoretical bounds scale like $C_1(1/n)^{2/(2+d)}$ and $C_2(\log n/n)^{2/(2+d)}$, which suggests that the true error curve should be well approximated by a power law of the form $R_n \sim C n^{-g(d)}$. Taking logarithms yields a linear model $\log R_n \sim -A \log n + \log C$. We therefore regress $(\log n_j, \log \hat{R}_{n_j})$ for a sequence of sample sizes $n_j$ covering the fractions $\{0.01, 0.02, 0.05, 0.1, \dots, 1.0\}$ of the available fitting pool. Each experiment is repeated three times, and the empirical values $\hat{R}_{n_j}$ used for the fit are the pointwise averages. The regression provides estimates $\hat{A}$ and $\hat{C}$, giving a fitted curve $\hat{R}_n^{\text{fit}} = \hat{C} n^{-\hat{A}}$.

We then use this fitted curve to select constants for the upper and lower theoretical bounds. For each $n_j$, we compute the 0.99 quantile of the empirical errors $\hat{R}_{n_j}$ across repetitions. The upper bound constant is chosen as the smallest value

for which the bound stays above these quantiles; the lower bound constant is chosen as the largest value for which the bound stays below them. This procedure yields stable and reproducible constants while preserving the theoretical scaling behavior. For more details and plots related to the logarithmic regression fitting, please look at Appendix G.2.

**Results:** We present three representative comparisons:

- **Cross-dimension** (Figure 3): curves for dSprites with intrinsic dimensions $d = 1, 2, 3, 4$, again using MMLS.

- **Cross-model** (Figure 4): comparison of MMLS and $\beta$-VAE on 4D dSprites.

- **Cross-dataset** (Figure 5): curves for sphere, torus, COIL-20 and dSprites using MMLS.

**Observations:** Across datasets and models, we find that the scaling implied by (Fefferman et al., 2018) is closer to the empirical error curves than the dimension-only rates of (Genovese et al., 2012). Here "closer" refers to the fitted log-log decay exponent, not to predictive constants or absolute curve height, since all theoretical constants are selected post hoc to envelope the empirical curves. This is consistent with our use of MMLS in a linear regime: we restrict MMLS to a local $d$-plane fit (PCA), and the nonasymptotic results of (Aamari & Levrard, 2019) suggest that such linear local fitting should exhibit an exponent close to $1/d$ in the low-noise setting. The $\beta$-VAE curve is comparatively flat, with a decay exponent close to 1/d or smaller. In this configuration, the $\beta$-VAE is not behaving like a high-order local fitter

on our controlled manifolds, which in turn motivates the kinds of controlled sweeps over architectures, $\beta$ values, and training budgets that the framework is designed to enable.

**Manifold analysis:** In the second application, we analyze how geometric properties of data manifolds evolve through the layers of a $\beta$-VAE using the dSprites (4D) and COIL-20 (3D) datasets. The purpose is to show that the benchmark can produce per-layer ground-truth geometric measurements that are otherwise difficult to obtain. For each layer we compute the manifold volume, integrated scalar curvature (with positive and negative parts separated), reach, and the average distance between class manifolds (Figure 6). To better illustrate the behavior of the image-based finite-difference estimates, and in particular the sensitivity of curvature to rasterization at fixed image resolution, we also analyze volume and curvature on a controlled ellipse example in Appendix F.

We observe that curvature systematically increases while reach decreases in deeper layers, indicating that the intermediate manifolds become progressively more intricate and approach self-intersections. At the same time, average distances between class manifolds increase toward the latent representation, suggesting that as semantic information strengthens, the network prioritizes class separation over preserving low-level visual transformations.

## 6. Discussion

We presented a framework for constructing dense manifolds and accurately estimating their geometric properties, enabling controlled studies of manifold fitting. We illustrated two use cases: assessing existing manifold fitting bounds via log-log scaling fits, and analyzing how data geometry evolves across layers of a $\beta$-VAE. Both revealed how geometric structure influences learning.

The main strength of this framework is its ability to provide ground-truth geometric quantities that are otherwise inaccessible or unreliable. Unlike real datasets, where assumptions cannot be verified, or simple analytic manifolds, which often lack representational richness, our synthetic constructions balance realism and control. This makes them well suited for probing theoretical assumptions, validating estimators, or stress-testing bounds under controlled perturbations.

Several limitations should nonetheless be emphasized:

- The framework can analyze only manifolds of low dimension (up to 4-5). It is therefore intended for benchmarking and understanding rather than for analyzing arbitrary datasets.

- It currently supports manifolds with simple topology, $[0, 1]^r \times (S^1)^s$. More general manifolds can still be handled by covering them with charts and applying the framework chart-wise.

- For rasterized image datasets, very fine transformation grids can introduce aliasing artifacts if image resolution is fixed. Grid refinement should therefore be paired with sufficient image resolution or regularization such as metric-tensor smoothing.

Future work could expand the framework or use it for further controlled experiments. Expanding the dataset suite with richer transformations, occlusions, and additional modalities (e.g., text, audio) would broaden applicability. Another possibility is to evaluate existing geometric estimators, e.g. curvature or reach, by comparing them against the more accurate finite-difference values on the test manifolds. The framework could also be used to study discriminative bounds, where classifiers fit functions on manifolds, and to quantify how geometric fidelity relates to generalization.

## Impact Statement

This paper presents work whose goal is to advance the understanding of machine learning methods and manifold analysis. While there are many potential societal consequences of our work, we do not expect immediate societal impact outside of the community on manifold learning.

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

## A. Bounds from Fitting a putative manifold to noisy data (Fefferman et al., 2018)

Here we briefly derive the sample complexity of the bound described in (Fefferman et al., 2018). There, they assume $M$ is a $\mathcal{C}^2$ $d$-dimensional manifold in $\mathbb{R}^D$ with reach at least $\tau$ and without boundary. The points are sampled i.i.d. from a measure $\mu$ on $M$ such that

$$0 < \rho_{\min} < \frac{d\mu}{d\lambda_M} < \rho_{\max} < \infty,$$

where $\lambda_M$ is the $d$-dimensional Hausdorff measure on $M$. Additionally, i.i.d. variables following a $D$-dimensional isotropic Gaussian of standard deviation $\sigma$ are added as noise to the points. Let $V = \lambda_M(M)$ denote the volume of $M$.

Based on Theorem 10 of (Fefferman et al., 2018), their algorithm constructs an output manifold $M_o$ with the same properties as $M$ such that, with high probability,

$$H(M_o, M) < Cd^7 \frac{r^2}{\tau}.$$

Here $r$ can be set to any value in $[\sqrt{\sigma\tau}D^{1/4}, \frac{\tau}{Cd^C}]$ which satisfies

$$\frac{n}{\log n} > \frac{CV}{\rho_{\min}\omega_d(r^2/\tau)^d}.$$

Solving for the term $\frac{r^2}{\tau}$ appearing in the Hausdorff bound, we get

$$\frac{r^2}{\tau} > \left(\frac{CV}{\rho_{\min}\omega_d}\right)^{\frac{1}{d}} \left(\frac{\log n}{n}\right)^{\frac{1}{d}}.$$

Moreover, from the lower end of the admissible range,

$$\frac{r^2}{\tau} > \sigma\sqrt{D}.$$

Combining these constraints yields

$$H(M_o, M) < Cd^7 \max\left\{\sigma\sqrt{D}, \left(\frac{CV}{\rho_{\min}\omega_d}\right)^{\frac{1}{d}} \left(\frac{\log n}{n}\right)^{\frac{1}{d}}\right\}.$$

Assuming small, or no, added noise, this simplifies to: for some constant $C_1$,

$$H(M_o, M) < C_1 \left(\frac{\log n}{n}\right)^{\frac{1}{d}}.$$

## B. Derivation of sample complexity for scalar curvature

Here we provide detailed derivations of the asymptotic sample complexity of the finite-difference methods we use.

### B.1. Useful lemmas for higher-order central differences

We define the central-difference operator in the form used throughout the paper. Note that the more common definition $f(x + \frac{h}{2}) - f(x - \frac{h}{2})$ is equivalent to ours after replacing $h$ by $h/2$.

**Definition B.1** (Central difference operator). For $h > 0$, define $\delta_h[f]$ by

$$\delta_h[f](x) = f(x + h) - f(x - h),$$

and define higher-order central differences recursively by

$$\delta_h^1[f] = \delta_h[f], \qquad \delta_h^{k+1}[f] = \delta_h[\delta_h^k[f]].$$

The following lemmas are standard, see, e.g., (Jordán, 1965), and we include short proofs for completeness.

**Lemma B.2** (Explicit expansion). *For any $k \geq 1$,*

$$\delta_h^k[f](x) = \sum_{i=0}^{k} (-1)^i \binom{k}{i} f(x + (k - 2i)h).$$

*Proof.* Let $E^a$ denote the shift operator $(E^a f)(x) = f(x + a)$. Then $\delta_h = E^h - E^{-h}$, hence

$$\delta_h^k = (E^h - E^{-h})^k = \sum_{i=0}^{k} (-1)^i \binom{k}{i} E^{(k-2i)h},$$

which yields the stated identity after applying both sides to $f$ and evaluating at $x$. $\quad\square$

**Lemma B.3** (Moment cancellations). *For integers $m \geq 0$ and $k \geq 1$, define*

$$S_m(k) := \sum_{i=0}^{k} (-1)^i \binom{k}{i} (k - 2i)^m.$$

*Then:*

- $S_m(k) = 0$ *for all* $m < k$,
- $S_k(k) = 2^k k!$,
- $S_{k+1}(k) = 0$.

*Proof.* Consider the polynomial $p_m(t) = t^m$. Using Lemma B.2 at $x = 0$ with $h = 1$ gives

$$\delta_1^k = \sum_{i=0}^{k} (-1)^i \binom{k}{i} (k - 2i)^m = S_m(k).$$

If $m < k$, then $\delta_1^k[p_m] = 0$ because each application of $\delta_1$ lowers polynomial degree by at least one, hence $S_m(k) = 0$. For $m = k$, the polynomial $\delta_1^k[p_k]$ is constant (degree 0), and its leading-term computation gives $\delta_1^k[p_k](t) = 2^k k!$, hence $S_k(k) = 2^k k!$. Finally, for $m = k + 1$, the polynomial $\delta_1^k[p_{k+1}]$ is linear and odd (since $\delta_1$ maps even polynomials to odd and vice versa), hence it vanishes at 0, giving $S_{k+1}(k) = 0$. $\quad\square$

**Lemma B.4** (Central differences approximate derivatives with $O(h^2)$). *Let $f : \mathbb{R} \to \mathbb{R}$ be $C^{k+2}$ in a neighborhood of $x$. Then*

$$\frac{1}{(2h)^k} \delta_h^k[f](x) = f^{(k)}(x) + O(h^2), \qquad h \to 0.$$

*Proof.* By Lemma B.2,

$$\delta_h^k[f](x) = \sum_{i=0}^{k} (-1)^i \binom{k}{i} f(x + (k - 2i)h).$$

Expand each term by Taylor's theorem with remainder:

$$f(x + (k - 2i)h) = \sum_{m=0}^{k+1} \frac{f^{(m)}(x)}{m!} (k - 2i)^m h^m + R_i, \qquad |R_i| \leq C |h|^{k+2}$$

for some constant $C$ depending on $\sup_{|t-x| \leq kh} |f^{(k+2)}(t)|$. Substituting into the sum yields

$$\delta_h^k[f](x) = \sum_{m=0}^{k+1} \frac{f^{(m)}(x)}{m!} h^m S_m(k) + \sum_{i=0}^{k} (-1)^i \binom{k}{i} R_i.$$

By Lemma B.3, all terms with $m < k$ vanish, the $m = k$ term equals

$$\frac{f^{(k)}(x)}{k!} h^k S_k(k) = \frac{f^{(k)}(x)}{k!} h^k (2^k k!) = (2h)^k f^{(k)}(x),$$

and the $m = k + 1$ term vanishes as well. The remainder satisfies

$$\left| \sum_{i=0}^{k} (-1)^i \binom{k}{i} R_i \right| \leq \left( \sum_{i=0}^{k} \binom{k}{i} \right) C|h|^{k+2} = 2^k C|h|^{k+2}.$$

Dividing by $(2h)^k$ proves the claim. $\qquad\square$

## B.2. Extension to partial and mixed derivatives

Let $f : \mathbb{R}^d \to \mathbb{R}$ and let $e_j$ denote the $j$-th standard basis vector. Define

$$(\delta_{h,j} f)(x) := f(x + he_j) - f(x - he_j), \qquad \delta_{h,j}^k := \underbrace{\delta_{h,j} \circ \cdots \circ \delta_{h,j}}_{k \text{ times}}.$$

Applying Lemma B.4 with all other coordinates held fixed yields:

**Lemma B.5.** *If $f$ is $\mathcal{C}^{k+2}$, then for each $j$,*

$$\frac{1}{(2h)^k} \delta_{h,j}^k [f](x) = \partial_j^k f(x) + O(h^2).$$

*More generally, for a multi-index $\alpha = (\alpha_1, \ldots, \alpha_d)$ with $|\alpha| = \sum_j \alpha_j$,*

$$\left( \prod_{j=1}^{d} (2h)^{-\alpha_j} \right) \left( \prod_{j=1}^{d} \delta_{h,j}^{\alpha_j} \right) [f](x) = \partial^\alpha f(x) + O(h^2),$$

*provided $f \in \mathcal{C}^{|\alpha|+2}$ and the stencil points remain in the domain.*

In practice, for non-cyclic coordinate directions we apply these central stencils only on interior grid points and crop a margin of width proportional to the stencil radius.

## B.3. Implication for scalar curvature

For an embedded chart $u : U \subset \mathbb{R}^d \to \mathbb{R}^D$, the scalar curvature computed via the Riemann tensor depends on derivatives of $u$ up to third order (through $g_{ij} = \langle \partial_i u, \partial_j u \rangle$, $\Gamma_{jk}^i$, and $\partial_\ell \Gamma_{jk}^i$). If $u \in \mathcal{C}^5$ on $U$ and the metric is uniformly non-degenerate, then:

- each partial derivative of $u$ up to order 3 is approximated with error $O(h^2)$ by Lemma B.5;

- the induced metric $g$, its inverse $g^{-1}$, Christoffel symbols, Riemann tensor, Ricci tensor, and scalar curvature are obtained from these derivatives via sums, products, and multiplication by $g^{-1}$, hence they inherit $O(h^2)$ error under the stated non-degeneracy condition.

Consequently, the pointwise scalar curvature estimator based on these finite differences converges at rate $O(h^2)$. For a grid with typical spacing $h \asymp n^{-1/d}$ (e.g. $n$ points in $d$ dimensions with comparable resolution), this translates to the sample-complexity scaling

$$\text{curvature error} = O(h^2) = O\left( n^{-2/d} \right),$$

up to constants depending on derivatives of $u$ and conditioning of the metric.

## C. Empirical comparison on scalar curvature estimation

As mentioned in the paper, we empirically compare our method with (Sritharan et al., 2021), which is one of the few scalar-curvature estimators with publicly available experiments and code. We evaluate on the following manifolds from their paper.

HYPERSPHERES $S^2, S^3, S^4$

We consider the unit hyperspheres $S^d \subset \mathbb{R}^{d+1}$ with parameters $\phi_1, \ldots, \phi_{d-1} \in [0.2\pi, 0.8\pi]$ and $\phi_d \in [0, 2\pi)$, using the standard hyperspherical parametrization

$$x_1 = \cos \phi_1,$$
$$x_i = \left( \prod_{j=1}^{i-1} \sin \phi_j \right) \cos \phi_i, \qquad i = 2, \ldots, d,$$
$$x_{d+1} = \left( \prod_{j=1}^{d-1} \sin \phi_j \right) \sin \phi_d.$$

The scalar curvature of $S^d$ is constant: $R(\phi_1, \ldots, \phi_d) = d(d-1)$.

HYPERBOLOID $H_2^2$

We consider a two-dimensional hyperboloid with parameters $\phi_1 \in [\sinh^{-1}(-1), \sinh^{-1}(1)]$ and $\phi_2 \in [0, 2\pi)$ and parametrization

$$x_1 = a_1 \sinh \phi_1,$$
$$x_2 = a_2 \cosh \phi_1 \sin \phi_2,$$
$$x_3 = a_3 \cosh \phi_1 \cos \phi_2.$$

The semi-axes are $(a_1, a_2, a_3) = (1, 2, 2)$. The range of $\phi_1$ is chosen so that the sampled patch has height 2 along the $x_1$-axis. The scalar curvature is

$$R(\phi_1, \phi_2) = -\frac{2}{\left( 5 \sinh^2 \phi_1 + 1 \right)^2}.$$

TORUS $T^2$

We consider a two-dimensional torus with parameters $\phi_1, \phi_2 \in [0, 2\pi)$ and parametrization

$$x_1 = (R + r \cos \phi_1) \cos \phi_2,$$
$$x_2 = (R + r \cos \phi_1) \sin \phi_2,$$
$$x_3 = r \sin \phi_1.$$

The radii are $(r, R) = (0.5, 2.5)$ and the scalar curvature is

$$R(\phi_1, \phi_2) = \frac{2 \cos \phi_1}{r(R + r \cos \phi_1)} = \frac{8 \cos \phi_1}{5 + \cos \phi_1}.$$

SAMPLING

For each target sample size $n \in \{2000, 4000, \ldots, 20000\}$ and intrinsic dimension $d$, we set $m = \lfloor n^{1/d} \rfloor$ and sample $m$ equidistant values for each parameter over its range to form a Cartesian grid. This yields $m^d = \lfloor n^{1/d} \rfloor^d$ points per manifold, hence the realized sample sizes do not generally match across different dimensions. For the hyperspheres we restrict $\phi_1, \ldots, \phi_{d-1}$ to a subset of $[0, \pi)$ to simplify the topology for the finite difference computations. Since both $S^d$ and the computations of both methods are $SO(d+1)$-invariant, applying them to two patches which are rotations of the current patch and cover $S^d$ would yield the same results.

METHODS CONFIGURATION

Our finite-difference estimator is run directly on the sampled grid, given the parameter ranges. To run the method of (Sritharan et al., 2021), we ported their MATLAB code from https://gitlab.com/hormozlab/ManifoldCurvature to Python and verified correctness by reproducing their reported results on uniformly sampled $S^2, S^3, S^5, S^7, H_2^2$, and $T^2$

using the hyperparameters from their Appendix D.3. We then applied both methods to the grid-sampled datasets above. Because the method of (Sritharan et al., 2021) is sensitive to its hyperparameters, we evaluated $50$ candidate values (per manifold) over ranges where the method converged and report the best mean squared error. These baseline scores are therefore mildly optimistic.

## D. Grid-Resolution Ablation for Image-Based Measures

The analytic manifolds provide ground truth for volume and curvature, but the image-based dSprites and COIL-20 manifolds do not. We therefore additionally test whether the estimated measures are stable when the grid resolution of the transformation parameters is increased. This ablation is not a substitute for ground truth, but it checks whether the quantities used in the paper are in a regime where further grid refinement changes them only moderately.

**Protocol.**  For each class we treat the corresponding image grid as a separate parametrized manifold. For dSprites, the parameters are horizontal translation, vertical translation, scale, and rotation. For COIL-20, the parameters are scale, in-plane rotation, and turntable view. We vary a target grid density $n$ around the value used in the main experiments; the final datasets in the paper use $n = 16$ values per transformation dimension, with additional buffer samples on non-cyclic dimensions where needed for finite differences. For each density and each class, we compute the pointwise volume element, scalar curvature, and local reach. The volume element is computed from the finite-difference metric $g_{ij} = \langle \partial_i F, \partial_j F \rangle$ as $\sqrt{\det g}$. Scalar curvature is computed from the same metric using central finite differences through the Christoffel symbols and Riemann tensor. Reach is estimated pointwise from the tangent-space formula

$$\rho(x) \approx \inf_{y \neq x} \frac{\|F(y) - F(x)\|^2}{2\, d(F(y) - F(x), T_x M)},$$

where tangent spaces are also estimated by finite differences.

The grids at two consecutive densities do not generally share the same parameter locations. Moreover, these are image manifolds at fixed spatial resolution, so increasing the parameter density can expose rasterization and interpolation artifacts rather than only reducing finite-difference error. For this reason we first interpolate the pointwise fields to a common quadrature grid and compare consecutive densities with a negative Sobolev norm instead of a pointwise $L^2$ norm. For a field difference $u = f_n - f_{n+1}$ on the common grid, we use

$$\|u\|_{H^{-1}}^2 = \sum_{\xi} (1 + \|\xi\|_2^2)^{-1} |\widehat{u}(\xi)|^2,$$

with exponent $p = 2$. This weak norm downweights high-frequency differences and is therefore less sensitive to small spatial shifts and rasterization noise. We report both the $H^{-1}$ norms of the fields and the relative consecutive errors

$$\frac{\|f_{n+1} - f_n\|_{H^{-1}}}{\|f_n\|_{H^{-1}}}.$$

**Observations.**  Figures 7-10 show that the volume element is the most stable of the three measures. Curvature has larger class-dependent variation and stronger high-frequency changes across grid densities, which is expected for rasterized image manifolds: at fixed image resolution, finer parameter grids increasingly probe interpolation and subpixel rasterization artifacts, and curvature amplifies such effects because it depends on further finite differences of the metric. Reach is also less stable, especially for a few dSprites classes where isolated spikes appear in the relative errors. This is consistent both with rasterization-induced local metric fluctuations and with the fact that the reach estimate is based on a minimum over pairs of points, making it sensitive to a small number of unstable witnesses. We study this rasterization effect more directly in Appendix F.

Overall, the ablation supports using the balanced image datasets at density $n = 16$ as a practical compromise between computational cost and estimator stability. The estimates are not fully converged in a strict asymptotic sense, and the plots should not be interpreted as ground-truth validation. They show, however, that the main trends are not caused by a single very coarse discretization choice.

Finally, increasing the parameter-grid density while keeping the image resolution fixed is not always beneficial. At high parameter density, neighboring images may differ mainly through subpixel rasterization and interpolation effects. Finite

**dSprites: Negative-Sobolev Norms**

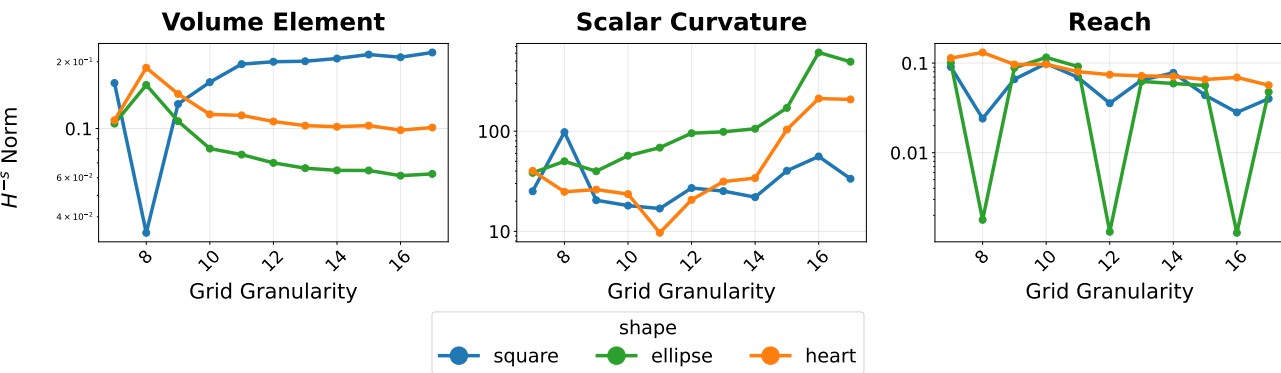

*Figure 7.* Negative-Sobolev norms of the pointwise volume element, scalar curvature, and reach for the dSprites grid-resolution ablation. Each curve corresponds to one class manifold.

**dSprites: Pairwise Relative Errors**

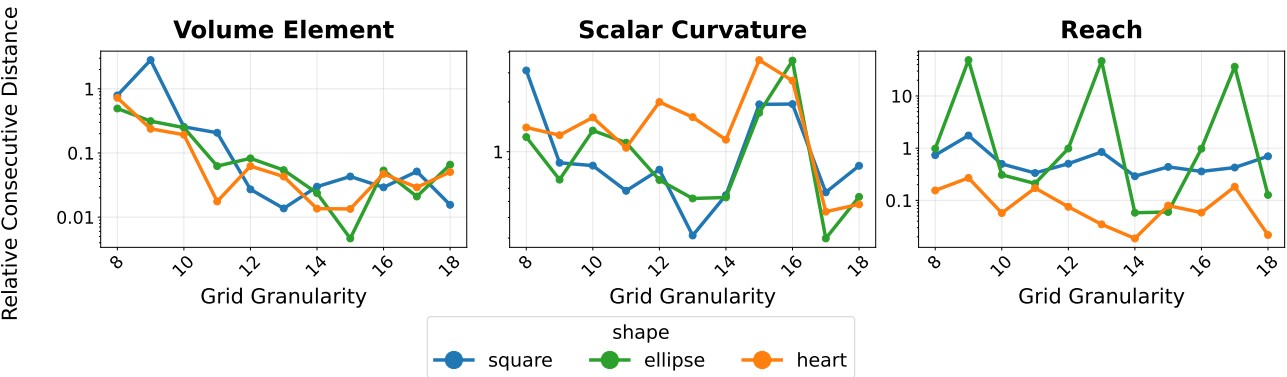

*Figure 8.* Relative consecutive negative-Sobolev errors for dSprites. The value at each plotted density compares the fields estimated at that density with the previous plotted density.

differences can amplify these small image-level changes, especially for metric derivatives and scalar curvature. We study this finite-image-resolution effect directly in Appendix F. Users generating new versions of these datasets should therefore compare resolutions with a weak metric such as the negative Sobolev norm, ideally increase the image resolution together with the parameter density, and consider smoothing the metric tensor. Metric-tensor smoothing is implemented in the released codebase and can reduce rasterization-induced high-frequency noise.

## E. Datasets

### E.1. dSprites

For the original dataset see (Matthey et al., 2017). Example images are shown in Fig. 11 and a 3D PCA projection in Fig. 12. In the PCA projection, class manifolds are strongly intertwined and partially envelop each other. The original dSprites rasterization exhibits visible aliasing artifacts, especially after rotation and rescaling on a $64 \times 64$ grid. To mitigate this, we generate each sprite at higher resolution and only downsample to $64 \times 64$ after applying the deformation transforms.

**Sprite construction (high-resolution).** We start from a binary mask on an empty background (zeros) on a canvas that is $s \times s$ with $s = 4 \cdot 64 = 256$ pixels. The sprite itself is centered and has initial size $4 \cdot 20 = 80$ pixels. We generate three shapes: a filled square, a filled ellipse (axes ratio 2:1), and a filled heart (from a parametric contour that is rasterized and filled). Intensities are stored as `float32`.

## COIL-20: Negative-Sobolev Norms

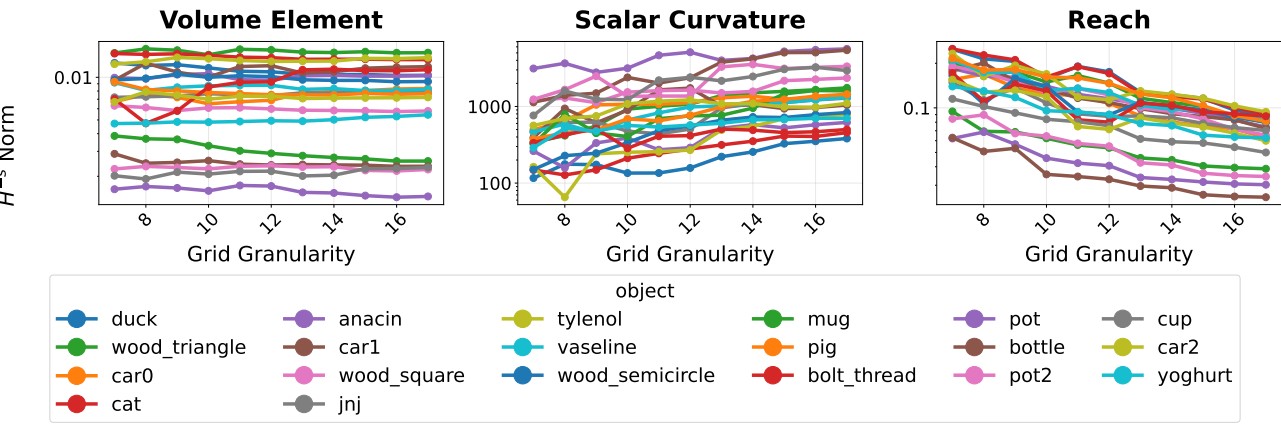

*Figure 9.* Negative-Sobolev norms of the pointwise volume element, scalar curvature, and reach for the COIL-20 grid-resolution ablation. Each curve corresponds to one object manifold.

## COIL-20: Pairwise Relative Errors

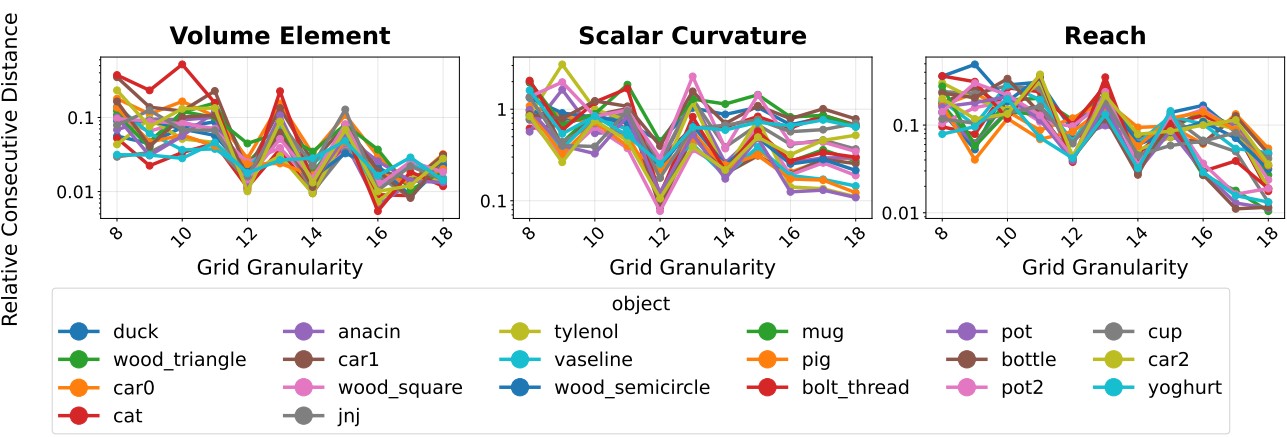

*Figure 10.* Relative consecutive negative-Sobolev errors for COIL-20. The value at each plotted density compares the fields estimated at that density with the previous plotted density.

**Transform pipeline.** For each shape we apply the following transforms in this order:

1. **Rotation (high-resolution):** rotate the $256 \times 256$ sprite by angles $\theta_j = 2\pi j / n_{\text{angles}}$, $j = 0, \ldots, n_{\text{angles}} - 1$ (i.e. uniformly spaced in $[0, 2\pi)$).

2. **Rescaling + downsampling:** for each scale factor $r$ (uniformly spaced in $[0.5, 1]$), we resample the rotated sprite to the target canvas size $64 \times 64$ using *area interpolation*. Conceptually, this step (i) sets the effective sprite size to $\lfloor 64\, r \rfloor$ pixels and (ii) undoes the initial $4\times$ upscaling by downsampling to the final resolution.

3. **Translation (final resolution):** translate the $64 \times 64$ image by integer offsets $(\Delta x, \Delta y)$ on the pixel grid, with $\Delta x, \Delta y \in \{-m, -m + \Delta, \ldots, m - \Delta\}$, where $m$ is the maximum translation and $\Delta$ is the translation stride. The background remains empty.

**Intensity normalization and smoothing.** After all transforms, we normalize intensities globally to $[-1, 1]$ (via min–max normalization over the generated grid). Optionally, we apply a Gaussian blur to the spatial axes (default $\sigma = 0.8$) to further

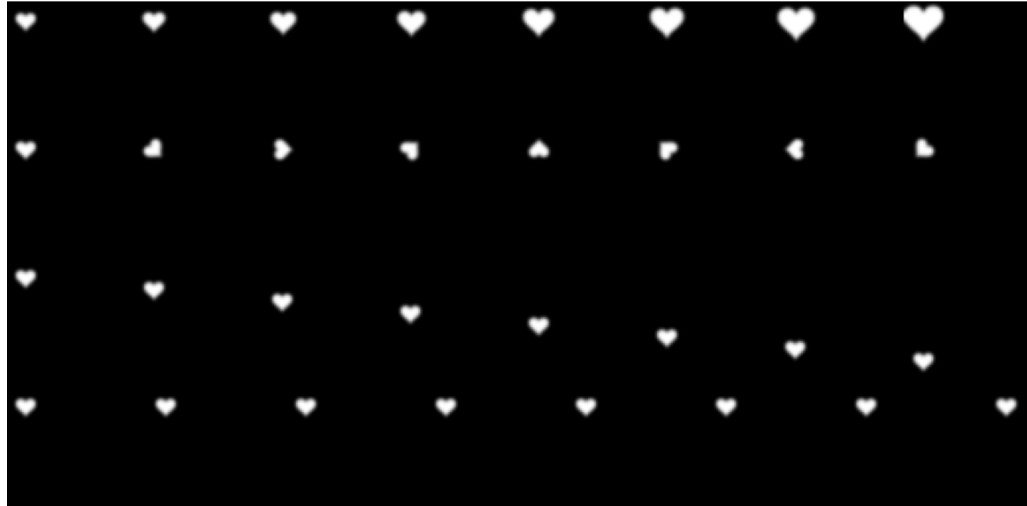

*Figure 11.* Example images of the heart class of dSprites.

suppress staircasing from discretization. Optionally, we also provide a standardized version where the dataset is centered and divided by its standard deviation.

**Output tensor.** The dataset is stored as a dense grid with shape

$$(n_{\text{shapes}}, n_{\text{scales}}, n_{\text{angles}}, n_{\Delta y}, n_{\Delta x}, 64, 64),$$

containing all combinations of factors.

**Parameters (balanced variant).** In our balanced version we use 3 shapes and 16 values per continuous factor (scale, orientation, and 2D position), yielding $3 \times 16^4 = 196{,}608$ images. For geometric estimation that requires local neighborhoods, we optionally add a bidirectional buffer of $b$ grid steps on the non-cyclic dimensions (scale and translations), i.e. we generate $16 + 2b$ values along those axes and treat only the central 16 as the evaluation domain.

- Image shape: $64 \times 64$

- Shape: 3 values (square, ellipse, heart)

- Scale: 16 values linearly spaced in $[0.5, 1]$ (optionally $16 + 2b$ with buffer)

- Orientation: 16 values in $[0, 2\pi)$

- Position X: 16 values on a uniform translation grid (optionally $16 + 2b$ with buffer)

- Position Y: 16 values on a uniform translation grid (optionally $16 + 2b$ with buffer)

### E.2. COIL-20

For the original dataset see (Nene et al., 1996). Example images are shown in Fig. 13 and a 3D PCA projection in Fig. 14. As in dSprites, the class manifolds are highly intertwined and partially envelop each other in low-dimensional projections.

**Base data (object turntable views).** We start from the preprocessed COIL-20 grayscale images (20 objects, 72 views per object, original resolution $128 \times 128$). Each view corresponds to a rotation of the object around the vertical axis on a turntable. To reduce redundancy, we optionally subsample the view index by a stride of 4, resulting in $72/4 = 18$ horizontal orientations per object. Pixel values are converted from $[0, 256)$ to $[0, 1]$ and stored as `float32`.

Classes

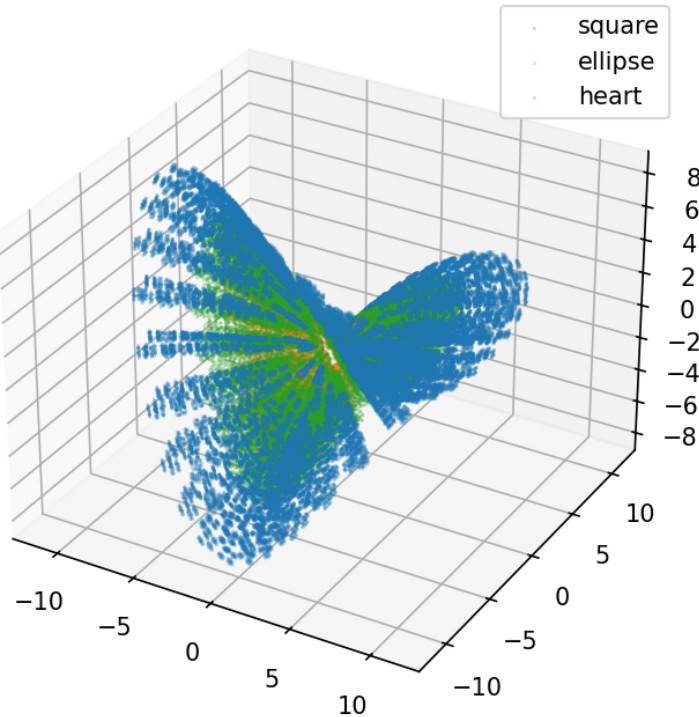

*Figure 12.* PCA projection of the dSprites dataset.

**Padding for deformation transforms.** To avoid cropping during in-plane rotations, each $128 \times 128$ image is embedded into a larger square canvas of size $s \times s$ with $s = 182 \approx 128\sqrt{2}$, by placing it centered on a zero background. This ensures that a subsequent $45°$ rotation fits within the padded frame up to discretization.

**Transform pipeline.** For each object and each subsampled turntable view, we apply the following transforms:

1. **In-plane rotation:** rotate the padded image by angles $\theta_j = 2\pi j / n_{\text{angles}}$, $j = 0, \ldots, n_{\text{angles}} - 1$ (uniformly spaced in $[0, 2\pi)$). This adds a second (independent) rotational factor beyond the original turntable viewpoint.

2. **Rescaling + cropping to target size:** for each scale factor $r$ (uniformly spaced in $[0.5, 1]$), we resample to the final resolution $64 \times 64$ using the same resizing routine as for dSprites. Operationally, we first set the effective crop size to $\lfloor 64\,r \rfloor$ and then map the result to the fixed $64 \times 64$ canvas, yielding a controlled scale factor while keeping the output size constant. To support finite-difference estimators that require neighboring grid points along non-cyclic axes, we optionally generate $n_{\text{sizes}} + 2b$ scales and use only the central $n_{\text{sizes}}$ for evaluation (buffer $b$).

**Intensity normalization.** After all transforms, intensities are mapped from $[0, 1]$ to $[-1, 1]$ by an affine rescaling. Optionally, we also provide a standardized version (global mean subtraction and division by standard deviation).

**Output tensor.** The resulting dense grid has shape

$$(20, \, n_{\text{views}}, \, n_{\text{angles}}, \, n_{\text{sizes}}, \, 64, \, 64),$$

where $n_{\text{views}} = 18$ after view subsampling, $n_{\text{angles}}$ is the number of added in-plane rotations, and $n_{\text{sizes}}$ is the number of scale values (optionally augmented with a buffer during generation).

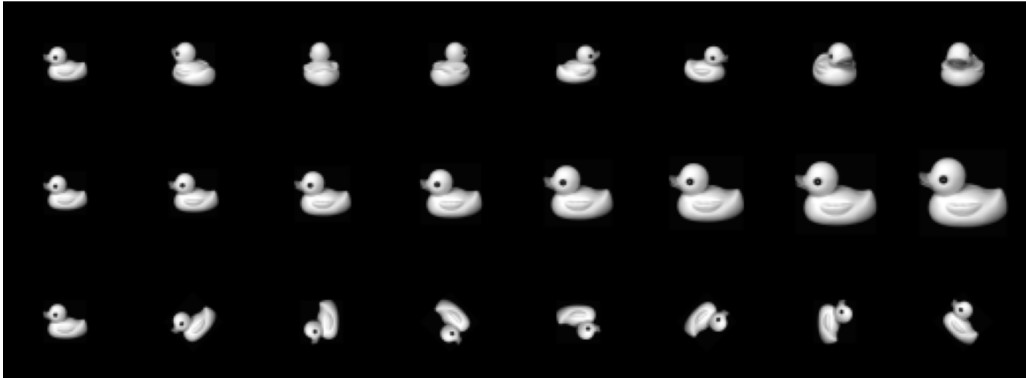

*Figure 13.* Example images of the duck class of COIL-20. Note the resizing and the additional in-plane rotation and scale transforms.

**Parameters.**

- Image shape: $64 \times 64$

- Objects: 20 objects from COIL-20

- Turntable view (horizontal orientation): 18 values (subsampled from 72 by stride 4)

- Added in-plane rotation: 16 values in $[0, 2\pi)$

- Scale: 16 values linearly spaced in $[0.5, 1]$ (optionally $16 + 2b$ with buffer)

- Total: 92,160 images ($20 \times 18 \times 16^2$), excluding any buffer-only samples

### E.3. Geometric Measures on Projections

Below we provide two 3D plots where the geometric measures are plotted on the heart class of dSprites (Fig. 15) and on the duck class of COIL-20 (Fig. 16). The absolute scalar curvature is the absolute value of the pointwise scalar curvature. In both datasets, the absolute scalar curvature exhibits smoother large-scale structure and aligns more clearly with the reach than the signed scalar curvature. The signed scalar curvature contains stronger local fluctuations, which should not be overinterpreted as purely geometric sign changes. For rasterized image manifolds, small interpolation and subpixel rasterization artifacts can be amplified by the finite-difference operations used to compute curvature. We isolate this effect in Appendix F.

## F. Rasterization Effects and Metric Smoothing on a Two-Dimensional Ellipse Manifold

The grid-resolution ablation in Appendix D compares the full image-based datasets across several densities. Here we isolate a simpler two-dimensional example in order to inspect the behavior of the finite-difference estimates more directly. We use only the ellipse class and restrict the transformation parameters to horizontal translation $x$ and in-plane rotation $\theta$. This gives a parametrized image manifold

$$F(x, \theta) \in \mathbb{R}^{64 \times 64},$$

where $x$ is non-cyclic and $\theta \in [0, 2\pi)$ is cyclic. Example images from this grid are shown in Fig. 17.

For each grid density we compute the Riemannian metric

$$g_{ij}(x, \theta) = \langle \partial_i F(x, \theta), \partial_j F(x, \theta) \rangle$$

using central finite differences. We then derive the pointwise volume element $\sqrt{\det g}$ and scalar curvature from this metric. In the plots below we show the diagonal metric components $g_{00}$ and $g_{11}$, the volume element, and the scalar curvature. The top row in each figure corresponds to the direct finite-difference estimate, while the bottom row uses the metric-smoothing procedure described below.

Classes

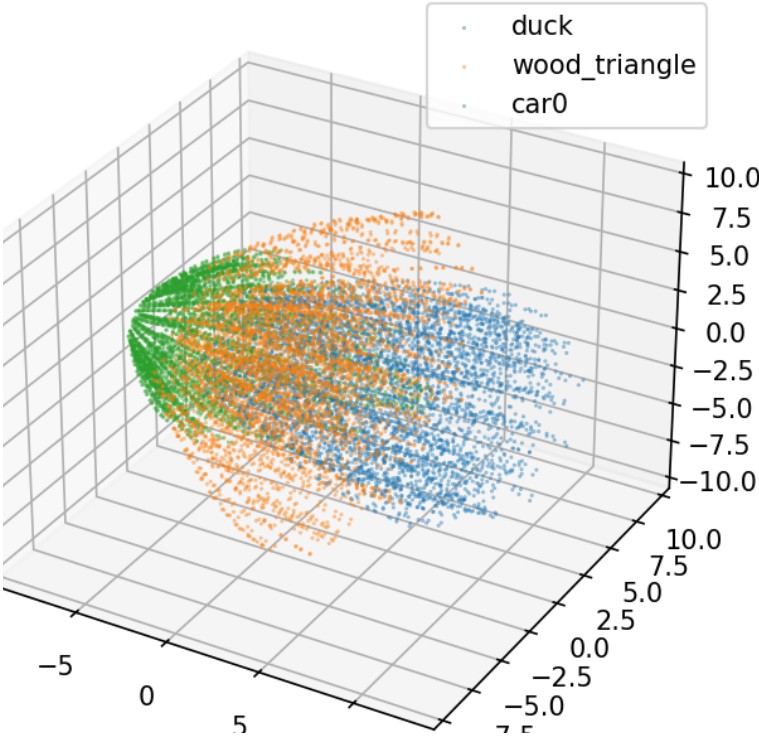

*Figure 14.* PCA projection of three classes of the COIL-20 dataset.

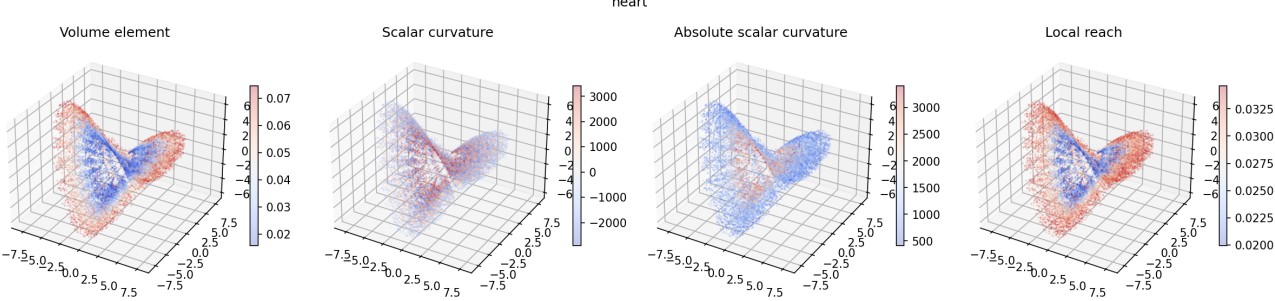

*Figure 15.* PCA projection of the dSprites heart class.

**Rasterization effects.** In a smooth continuous image model, increasing the grid density would reduce the finite-difference step size and should improve derivative estimates. For rasterized images at fixed spatial resolution, however, this is not guaranteed. When the parameter grid becomes very fine while the images remain $64 \times 64$, neighboring samples may differ mainly through interpolation and rasterization artifacts. The finite-difference quotient can then amplify small subpixel changes, especially for curvature, which uses several derivative operations. This effect is visible in the unsmoothed rows of Figs. 18–21: high-frequency oscillations appear as the transformation grid becomes finer, even though the underlying geometric transformation is simple.

**Metric-tensor smoothing.** A simple mitigation is to smooth the estimated metric tensor before computing the derived quantities. In our implementation, the finite-difference metric is first computed componentwise and then filtered over the parameter grid,

$$\widetilde{g}_{ij} = G_\sigma * g_{ij},$$

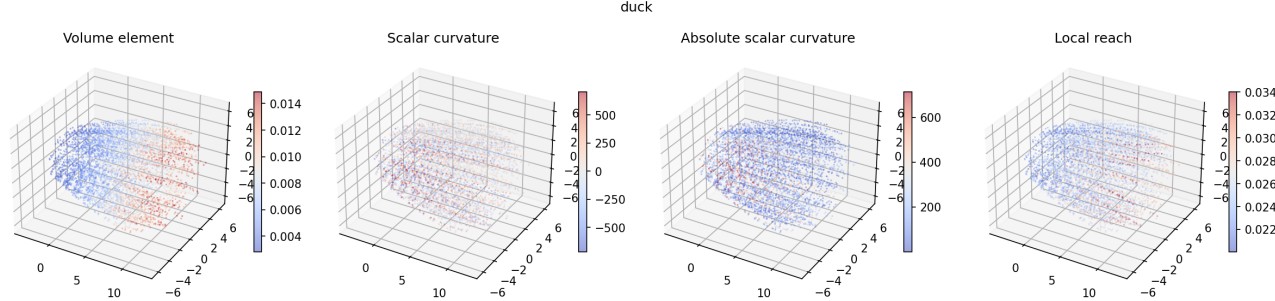

*Figure 16.* PCA projection of the COIL-20 duck class.

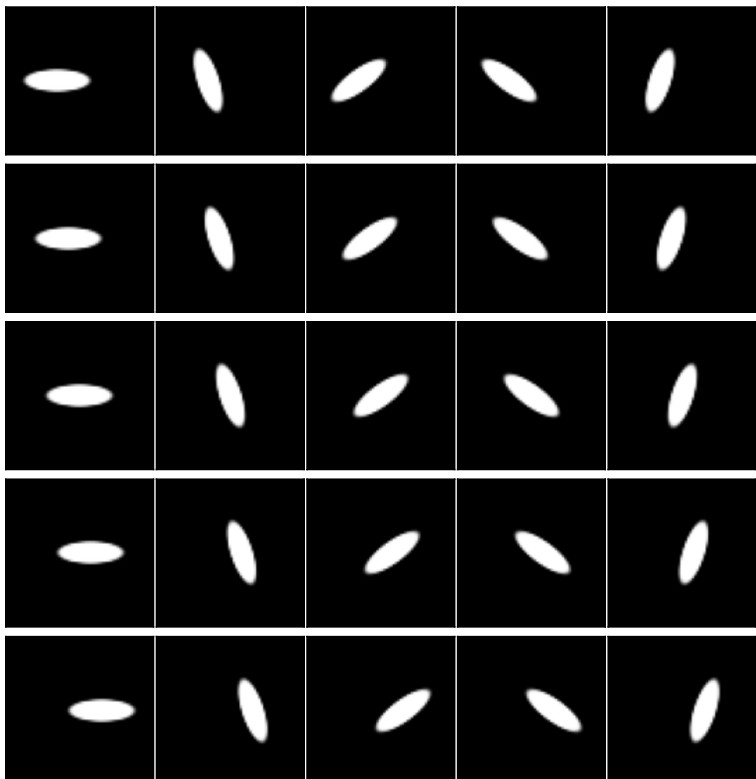

*Figure 17.* Example images from the two-dimensional ellipse grid generated by varying horizontal translation $x$ and rotation $\theta$.

where $G_\sigma$ is a separable Gaussian kernel. We use reflection padding along the non-cyclic $x$ axis and wrapped padding along the cyclic $\theta$ axis. The implementation is done in PyTorch and is available in the released codebase. The volume element and scalar curvature are then computed from $\widetilde{g}$ instead of $g$.

This smoothing is meaningful when the parameter grid is already fine enough that neighboring metric values should not change abruptly under the underlying continuous transformation. Under this assumption, high-frequency oscillations are more likely to come from rasterization and interpolation than from the manifold geometry itself. The smoothed rows in Figs. 18–21 show that this procedure reduces much of the high-frequency variation, especially for scalar curvature. At the same time, smoothing can also remove genuine sharp variation if used too aggressively, so we treat it as a numerical stabilization tool rather than as part of the definition of the measures.

The main experiments in the paper use the unsmoothed estimates. This is acceptable for the reported datasets because the grid density used there is moderate, and the convergence ablation in Appendix D suggests that the estimates are already in a

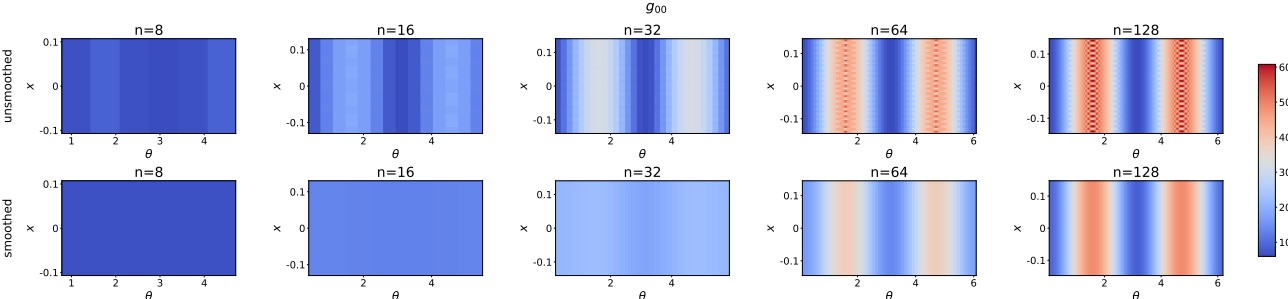

*Figure 18.* Estimated metric component $g_{00}$ for the ellipse grid. The top row uses the direct finite-difference metric; the bottom row smooths the metric tensor before computing derived quantities.

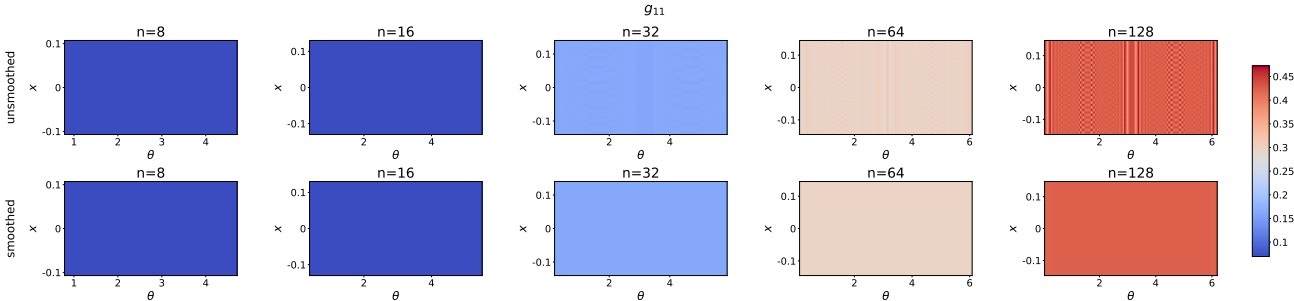

*Figure 19.* Estimated metric component $g_{11}$ for the ellipse grid, with the same layout as Fig. 18.

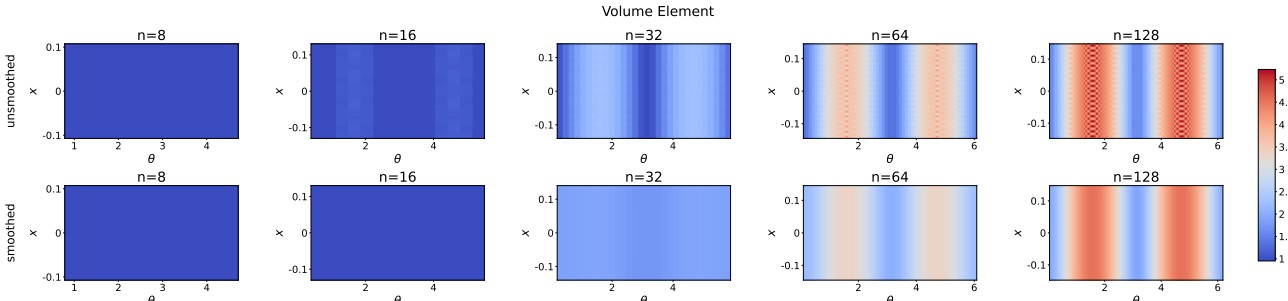

*Figure 20.* Estimated volume element on the ellipse grid. Rasterization effects are milder than for curvature but are still visible at high grid densities.

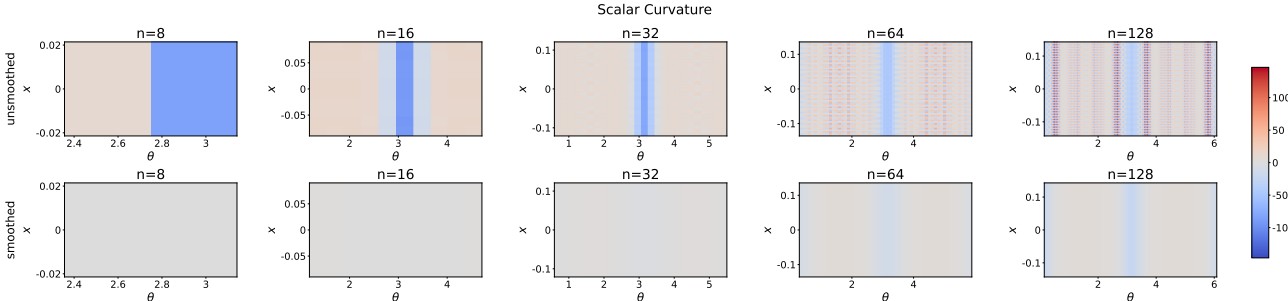

*Figure 21.* Estimated scalar curvature on the ellipse grid. Curvature is the most sensitive of the displayed quantities because it depends on further finite differences of the metric.

useful range at the chosen density. For users generating denser versions of the datasets, however, we recommend checking the sensitivity to rasterization and considering metric smoothing, especially for curvature.

**Projection view.** Figure 22 shows the same example through a two-dimensional UMAP (McInnes et al., 2018) projection of the image samples. The projection is used only for visualization; the measures themselves are computed on the original transformation grid. The same UMAP coordinates are used for the unsmoothed and smoothed rows, so differences between rows reflect only the change in the estimated measures. The projection confirms the same qualitative picture: smoothing reduces local high-frequency fluctuations while preserving the broader variation of the metric and volume-related quantities.

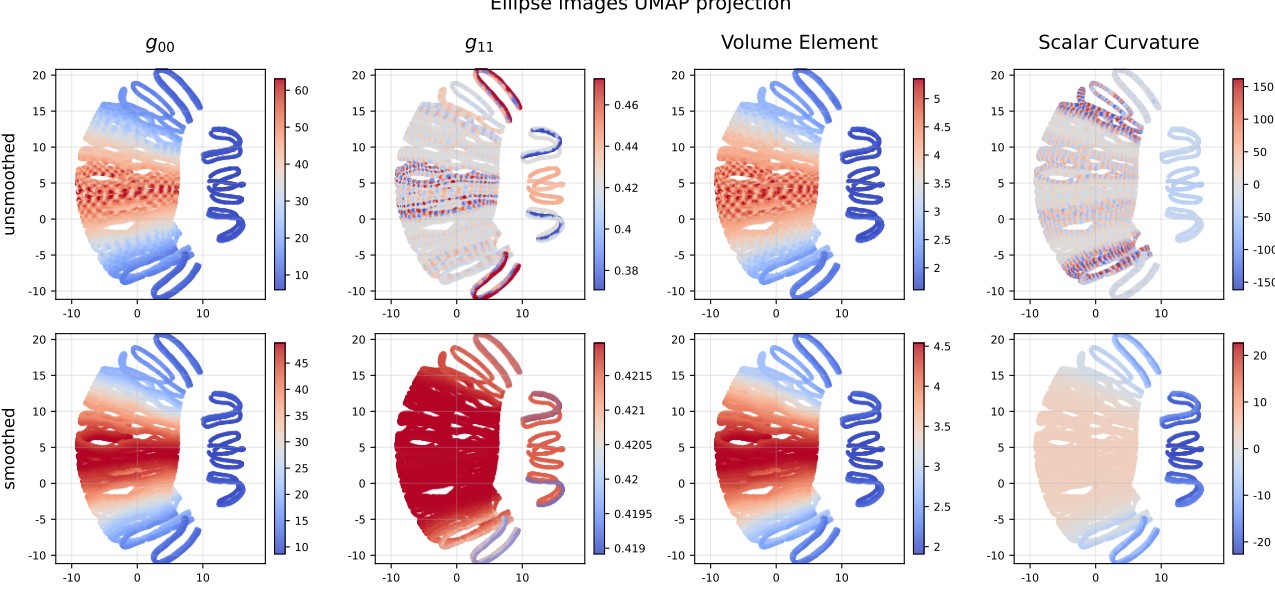

*Figure 22.* UMAP projection of the ellipse images colored by the estimated measures. The top row uses the direct finite-difference estimates and the bottom row uses metric smoothing. The UMAP coordinates are identical across rows.

**Practical warning.** The data manifolds considered here are manifolds of rasterized images, not continuous shapes. At fixed image resolution, increasing the parameter-grid density eventually probes the discretization of the image formation process rather than only the underlying continuous transformation. This is a substantial issue in its own right and we leave a systematic study of rasterization effects to future work. In practice, one should be careful when generating new versions of these datasets with very high parameter density but unchanged image resolution. A safer procedure is to increase the image resolution together with the parameter density, compare estimates across densities using a weak metric such as the negative Sobolev norm, and optionally smooth the metric tensor to suppress high-frequency rasterization noise.

# G. Manifold Fitting

Here we provide details on the way we setup and run the experiments for the manifold fitting use case of our framework. This included details on the datasets used, the fitting models and the way we estimate the bound curves.

## G.1. Fitting methods

THE FITTING PROCESS AND DATA SELECTION

Most manifold-fitting methods estimate local structures on $M$, such as tangent $d$-planes, tubular neighborhoods, or charts, from a finite fitting set. These local structures define a reconstructed manifold $\widehat{M}$ together with a projection or reconstruction map onto $\widehat{M}$. Evaluating the fit ideally requires computing the Hausdorff distance $H(M, \widehat{M})$. A direct numerical approximation would require dense, uniformly distributed point sets on both $M$ and $\widehat{M}$, together with exhaustive nearest-neighbor searches, which is computationally expensive.

We therefore separate the data used for fitting from the data used for evaluation. Throughout this section, $X$ denotes the fitting or training set, while $Y$ denotes a dense evaluation/reference set sampled from the ground-truth manifold. After fitting on $X$, we evaluate the approximation on $Y$. For each $y \in Y$, let $\widehat{y}$ be its projection or reconstruction on $\widehat{M}$. The projection error $\|y - \widehat{y}\|_2$ serves as a surrogate for $d_{\ell^2}(y, \widehat{M})$, and the directed Hausdorff term $\sup_{y \in M} d_{\ell^2}(y, \widehat{M})$ is approximated by the maximum error over the dense reference set $Y$. The reverse direction and the relation to the full symmetric Hausdorff distance are discussed below.

**Toy manifolds.** Our procedure is:

- Construct a dense, approximately uniform evaluation set $Y = \{y_i\}_{i \leq n_{\text{test}}} \subset M$.

- Uniformly sample $n$ fitting points $X = \{x_i\}_{i \leq n} \subset M$.

- Fit the chosen manifold-fitting method on $X$.

- Project the evaluation points $Y$ to $\widehat{Y} = \{\widehat{y}_i\}_{i \leq n_{\text{test}}}$ and compute the pointwise errors $\|y_i - \widehat{y}_i\|_2$ and their maximum.

The dense evaluation set $Y$ is generated either directly via a grid in a closed-form parametrization or by drawing a large uniform sample $S = \{s_i\}_{i \leq N} \subset M$ and selecting $n_{\text{test}}$ centroids $C = \{c_i\} \subset M$ that minimize the Sinkhorn loss between $C$ and $S$. All uniform sampling steps rely on closed-form sampling formulas tailored to each manifold.

**Image manifolds.** The procedure mirrors the analytic case:

- Select a dense evaluation subset $Y = \{y_i\}_{i \leq n_{\text{test}}} \subset X_G$ from the grid $X_G$.

- Sample a fitting set $X = \{x_i\}_{i \leq n} \subset X_G \setminus Y$.

- Fit the manifold-fitting method on $X$.

- Project or reconstruct the evaluation points $Y$ to $\widehat{Y} = \{\widehat{y}_i\}_{i \leq n_{\text{test}}}$ and compute $\|y_i - \widehat{y}_i\|_2$ and their maximum.

The dense evaluation subset $Y$ is selected by a maximal-distance subsampling algorithm. Sampling of the fitting set $X$ uses a discrete distribution obtained by normalizing the per-point volume element on the grid. Projection is geometric for MMLS and is given by reconstruction for the $\beta$-VAE.

MANIFOLD MOVING LEAST SQUARES (MMLS)

MMLS (Sober & Levin, 2020) is a projection-based manifold fitting method. Given sample points $Y = \{y_i\}_{i \leq N} \subset \mathbb{R}^D$ drawn from an unknown manifold $M$, the algorithm estimates a fitted manifold $\widehat{M}$ and provides a projection operator that maps any nearby point $p$ to $\widehat{M}$. A key ingredient is a kernel $\theta$ that assigns similarity weights based on point–point distances, enabling smooth local approximations.

The procedure consists of two stages:

- **Local affine fitting.** For a query point $p \in \mathbb{R}^D$ close to $M$, find an affine subspace $H$ and a point $q \in B_D(p, \frac{\tau}{2})$ that minimize

$$\sum_{i \leq N} d_{\ell^2}(y_i, H)^2 \, \theta(\|y_i - q\|_2),$$

subject to $p - q \perp H$, where $\tau$ denotes the reach of $M$. This step provides a locally estimated tangent space.

- **Local polynomial reconstruction.** With $q$ and $H$ fixed, fit a multivariate polynomial $g : \mathbb{R}^d \to \mathbb{R}^D$ by solving

$$\sum_{i \leq N} \|g(\text{proj}_H(y_i)) - y_i\|_2^2 \, \theta(\|y_i - q\|_2),$$

where $\text{proj}_H(y_i)$ is the orthogonal projection onto $H$. This serves as a local analogue of the exponential map, taking coordinates in the tangent space $T_q\widehat{M}$ and mapping them to $\widehat{M}$. The final projection of $p$ is then $g(p)$.

The method originates from the Moving Least Squares framework for hypersurfaces (Levin, 2003) and was later extended to arbitrary embedded manifolds (Sober & Levin, 2020). Other techniques exist (e.g., (Zhang & Zha, 2004) and the constructions used in (Fefferman et al., 2016; 2018; Genovese et al., 2012)), but these are typically more intricate or tailored to dimensionality reduction rather than direct manifold fitting.

**Implementation for our experiments.**  We apply MMLS using the uniformly sampled fitting set $X$ as the reference set and the dense evaluation set $Y$ as queries. Each $y \in Y$ is projected onto $\widehat{M}$ as follows:

- Identify the $k$ nearest neighbors $N_y = \{x_{i_1}, \ldots, x_{i_k}\} \subseteq X$.

- Compute distances $\|y - x_{i_m}\|_2$ for all neighbors and assign weights via an isotropic Gaussian kernel with bandwidth $\sigma$:

$$w_m = \exp\left(-\left(\frac{\|y - x_{i_m}\|_2}{2\sigma}\right)^2\right).$$

- Estimate $q$ as the weighted average of $N_y$. Estimate the affine space $H$ by performing weighted PCA on $N_y$ with weights $w_m$.

- Project $y$ orthogonally onto $H$. In place of the full polynomial stage in (Sober & Levin, 2020), we use a degree-1 local linear approximation.

**Hyperparameters.**

- $k = 5$ for toy manifolds and $k = 2^{d+1}$ for image manifolds.

- Gaussian kernel bandwidth: $\sigma = 1.0$.

$\beta$-VAE

We use the reference implementation from (WonKwang Lee, 2018), which provides two standard architectures: model **B** from (Higgins et al., 2017) and model **H** from (Burgess et al., 2018). Both operate on $64 \times 64$ images and differ mainly in channel width and bottleneck design.

**Model B.**

Encoder: Conv $32 \times 4 \times 4$ (stride 2), $32 \times 4 \times 4$ (stride 2), $32 \times 4 \times 4$ (stride 2), $32 \times 4 \times 4$ (stride 2), FC $256 \rightarrow 256 \rightarrow 2z_d$.

Decoder: FC $z_d \rightarrow 256 \rightarrow 256 \rightarrow 512$, followed by Deconv layers mirroring the encoder (stride 2).

All layers use ReLU activations.

**Model H.**  Encoder: Conv $32 \times 4 \times 4$ (stride 2), $32 \times 4 \times 4$ (stride 2), $64 \times 4 \times 4$ (stride 2), $64 \times 4 \times 4$ (stride 2), $256 \times 4 \times 4$ (stride 1), FC $2z_d$.

Decoder: FC $z_d \rightarrow 256$, followed by Deconv layers reversing the encoder (strides 1 and 2).

All layers use ReLU activations.

**Objective.**  Model **H** uses the standard ELBO with a strengthened KL term:

$$\mathcal{L} = \text{recon} + \beta\, D_{\text{KL}}, \qquad \beta > 1.$$

Model **B** follows the capacity-controlled formulation:

$$\mathcal{L} = \text{recon} + \gamma\, |D_{\text{KL}} - C|,$$

where $C$ is annealed linearly from 0 to $C_{\max}$.

**Hyperparameters.**

- $\beta = 4$, $\gamma = 100$, $C_{\max} = 20$.

- Learning rate: $5 \times 10^{-4}$ (dSprites), $10^{-4}$ (COIL-20).

- Batch size: 64.

- Latent dimension: $z_d = 10$.

- Optimizer: Adam with $\beta_1 = 0.9$, $\beta_2 = 0.999$.

- Architecture choice: model B for dSprites, model H for COIL-20.

- Training budget: up to $10^6$ iterations or $10^5$ epochs.

- Early stopping when training loss does not improve for $0.5\%$ of max epochs.

These settings closely follow (WonKwang Lee, 2018), with minimal tuning for stable convergence. After training on the fitting set $X$, the decoder is used to reconstruct the evaluation points $Y$.

### G.2. Bound Curves

TAKEAWAYS ON CONSTANTS IN THE GENOVESE ET AL. BOUNDS

The lower and upper bounds of (Genovese et al., 2012) are primarily useful for their *rates* in $n$ and their dependence on intrinsic dimension $d$. While their proofs in principle yield explicit constants, those constants become unwieldy and are not practically interpretable for high-dimensional ambient data.

**Lower bound.** The lower bound is obtained via a two-point testing construction (Le Cam's method): two manifolds $M_0, M_1$ are constructed at Hausdorff separation $\gamma$ while keeping the induced distributions close, yielding the rate $n^{-2/(d+2)}$. The associated constant depends on geometric parameters (e.g. reach and noise radius) and on the specifics of the construction (choices of bump geometry and localization). As a result, explicitly tracking the constant provides little insight beyond confirming the scaling.

**Upper bound and why constants become intractable.** The upper bound is proved by constructing a (theoretical) estimator based on maximum likelihood and controlling its error via entropy bounds. A key step is bounding the Hausdorff covering number of the manifold class: Theorem 9 bounds

$$N(\gamma, \mathcal{M}, H) \leq \kappa_2(\kappa, d, D) \exp\left(\kappa_3(\kappa, d, D) \gamma^{-d/2}\right),$$

where already the prefactor satisfies

$$\kappa_2(\kappa, d, D) = \binom{D}{d}\left(\frac{c_2}{\kappa}\right)^D.$$

Thus, combinatorial terms (the factor $\binom{D}{d}$) and exponential dependence on ambient dimension enter *before* subsequent steps (bracketing, MLE concentration, and conversion from Hellinger to Hausdorff distance), making the resulting explicit constants extremely large for realistic $D$.

**Practical use in this paper.** Accordingly, we treat these results as guidance on *scaling* and on which geometric parameters matter, and we estimate envelope constants empirically by aligning the theoretical rates to observed error curves.

ESTIMATING THE BOUNDS' CONSTANTS

Because directly computing the constants in the bounds is impractical, we select them based on the empirical error curves. The idea is to simply place the upper bounds so that they just touch the empirical curve from above and the lower bounds so that they just touch the empirical curve from below. To get more reliable results, we first fit a curve on the empirical curve and use this to place the bounding curve. The selection of the curves family for the empirical curve is based on the

observation that the formulas related to bounds depend exponentially on the dimension and multiplicatively on the constant for example looking at the Genovese et al. upper and lower bounds:

$$C_1 \left(\frac{1}{n}\right)^{\frac{2}{2+d}} \leq R_n(\mathcal{Q}) \leq C_2 \left(\frac{\log n}{n}\right)^{\frac{2}{2+d}}. \tag{17}$$

It is reasonable thus to assume that the real error curve can be well approximated by a formula of the form:

$R_n \sim C(\frac{1}{n})^{g(d)}$,

which can be written in terms of logarithms as:

$\log R_n \sim -g(d) \, \log n + \log C = -A \, \log n + \log C$.

With this assumption, we use the logarithms of the empirical curve values $\{(\log n_j, \log \hat{R}_{n_j})\}_{j \in J}$, where $\{n_j \mid j \in J\} \subseteq [0, N]$ is an increasing sequence of number of samples of the total $N$ dataset points, to fit the above regression. This way we get estimates $\hat{A}, \hat{C}$ of $A$ and $C$. Then we get the resulting fitted empirical curve:

$$\hat{R}_n^{fit} = \hat{C} \left(\frac{1}{n}\right)^{\hat{A}}.$$

Finally, for each sample size $n_j$, we compute the 0.99 quantile of the empirical errors across repetitions. The upper-bound constant is chosen as the smallest value for which the theoretical curve lies above these quantiles, and the lower-bound constant is chosen as the largest value for which the theoretical curve lies below them. In our experiments, the sample sizes correspond to the fractions

$$[0.01, 0.02, 0.05, 0.1, \ldots, 1.0]$$

of the dataset of $N$ points. Each percentage is repeated three times, and the empirical curve used here is their pointwise average.

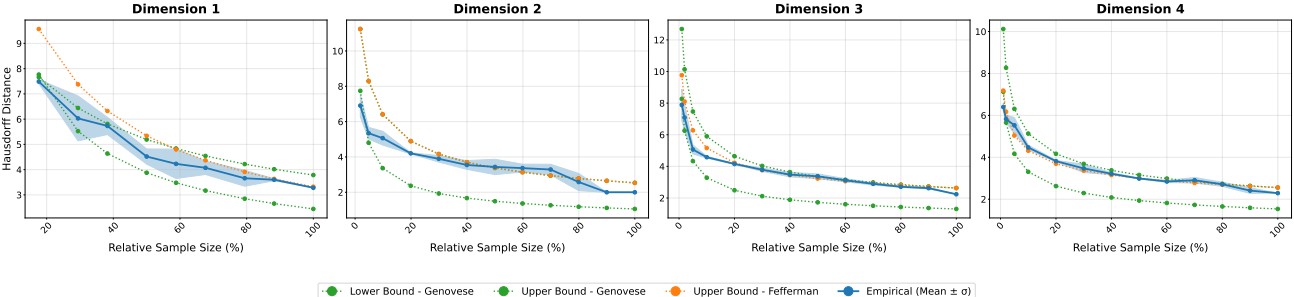

*Figure 23.* Fitting bounds on dSprites for different dimensions utilizing MMLS.

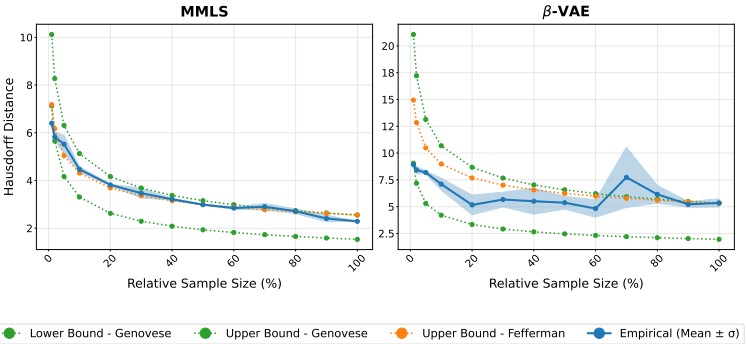

*Figure 24.* Fitting bounds for MMLS & $\beta$-VAE on dSprites (4D)

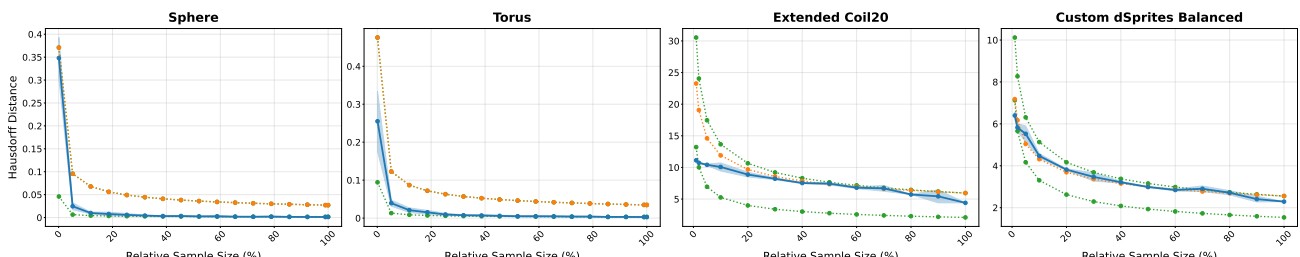

*Figure 25.* Fitting bounds for MMLS on from left to right Sphere, Torus, COIL-20 and dSprites.

ANALYSIS OF THE FITTED EMPIRICAL CURVES

To provide a better understanding on the way we select the upper and lower bounds, we display a version of figures 5, 4 and 3 including the fitted empirical curve as a blue dotted line in figures 25, 24 and 23.

Furthermore, we include figures 28, 27 and 26 with the regression line on the logarithmic values of the empirical curves, residual plots on the fitted values, QQ plots and also the values of $R^2$. Those are again available for all three ablations present in the main paper. Most fits look reasonably good, with the exception of the $\beta$-VAE plot, whose lower $R^2$ indicates that a single power law describes the empirical curve less well.

To assess whether the empirical curves are well described by the fitted power laws, Table 2 reports the $R^2$ values of the logarithmic regressions, the theoretical decay exponents used for the curves from (Genovese et al., 2012) and (Fefferman et al., 2018), and the fitted empirical exponent $\hat{A}$ in $\hat{R}_n^{\text{fit}} = \hat{C}n^{-\hat{A}}$. Most settings are well approximated by a single power law. The main exception is the $\beta$-VAE cross-model experiment, where the substantially lower value $R^2 = 0.62$ matches the fitting issues observed in Appendix G.2.

*Table 2.* Summary of the logarithmic power-law fits used for the bound comparisons. For each representative sweep, we report the regression quality $R^2$, the theoretical decay exponents used for the Genovese and Fefferman curves, and the empirically fitted exponent $\hat{A}$. The low $R^2$ value for the $\beta$-VAE setting is highlighted in bold.

| Setting | $R^2$ | Gen. | Feff. | Fit $\hat{A}$ |
|---|---|---|---|---|
| *Cross-dimension: dSprites, MMLS* | | | | |
| $d = 1$ | 0.99 | 0.67 | 1.00 | 0.48 |
| $d = 2$ | 0.85 | 0.50 | 0.50 | 0.27 |
| $d = 3$ | 0.97 | 0.40 | 0.33 | 0.25 |
| $d = 4$ | 0.96 | 0.33 | 0.25 | 0.22 |
| *Cross-model: 4D dSprites* | | | | |
| MMLS | 0.96 | 0.33 | 0.25 | 0.22 |
| $\beta$-VAE | **0.62** | 0.33 | 0.25 | 0.11 |
| *Cross-dataset: MMLS* | | | | |
| Sphere | 0.99 | 0.50 | 0.50 | 0.91 |
| Torus | 0.98 | 0.50 | 0.50 | 0.82 |
| COIL-20 | 0.79 | 0.40 | 0.33 | 0.17 |
| dSprites | 0.96 | 0.33 | 0.25 | 0.22 |

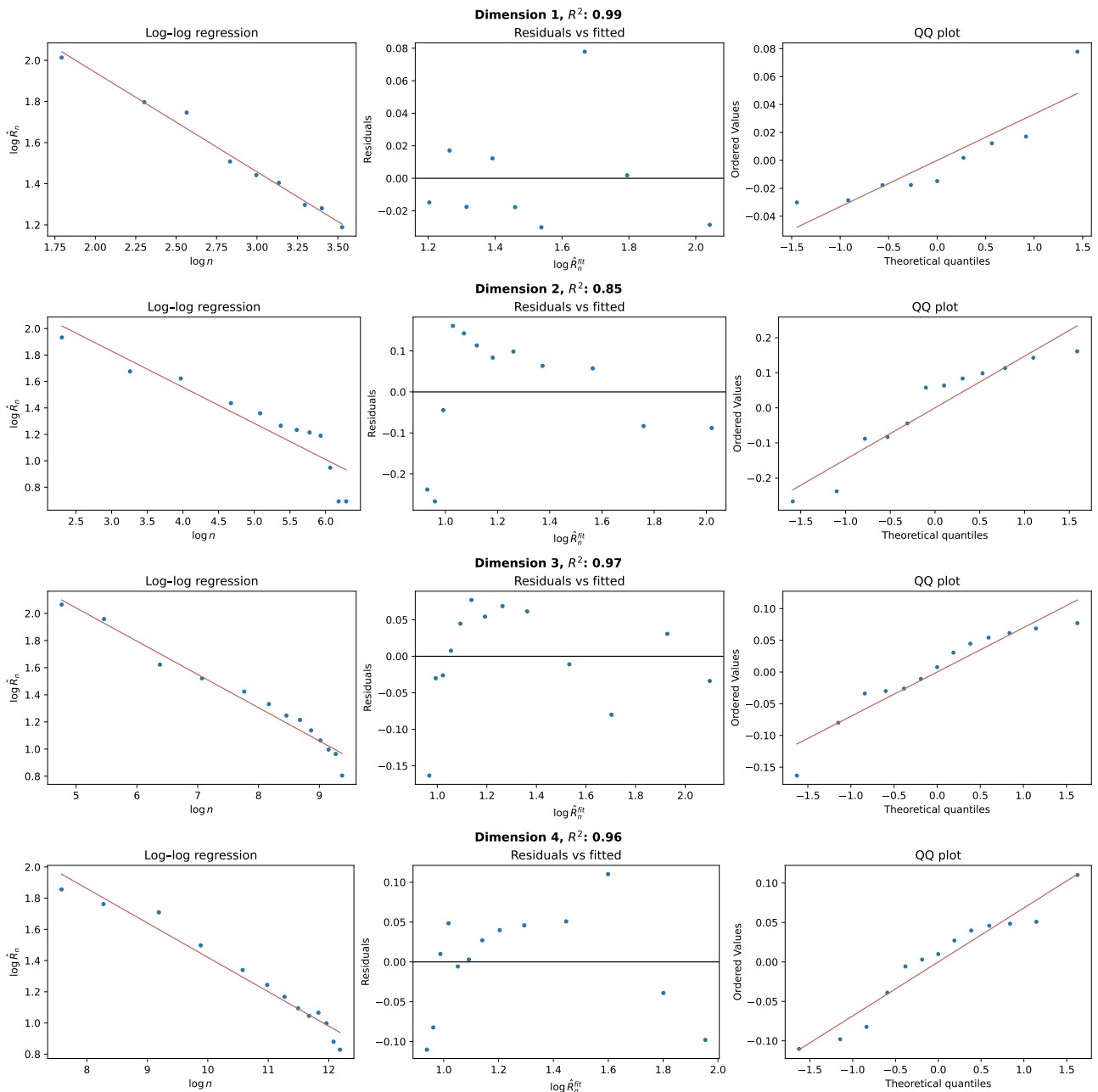

*Figure 26.* Regression evaluation on dSprites for different dimensions utilizing MMLS.

### Pointwise proxy for the Hausdorff distance

In the manifold-fitting experiments, computing the full symmetric Hausdorff distance at every sample size and repetition would require repeated nearest-neighbor searches between the original samples and their projections onto the fitted manifold. To reduce this cost, we use the pointwise proxy

$$d_{\mathrm{pw}}(X, \widehat{X}) = \max_i \|x_i - \widehat{x}_i\|,$$

where $x_i$ is an original sample and $\widehat{x}_i$ is its projected point. For the corresponding finite point clouds $X = \{x_i\}$ and $\widehat{X} = \{\widehat{x}_i\}$, this quantity upper-bounds the symmetric Hausdorff distance, since each point can always be matched to its paired counterpart. It can therefore overestimate the true Hausdorff distance, but should be a reasonable proxy when the

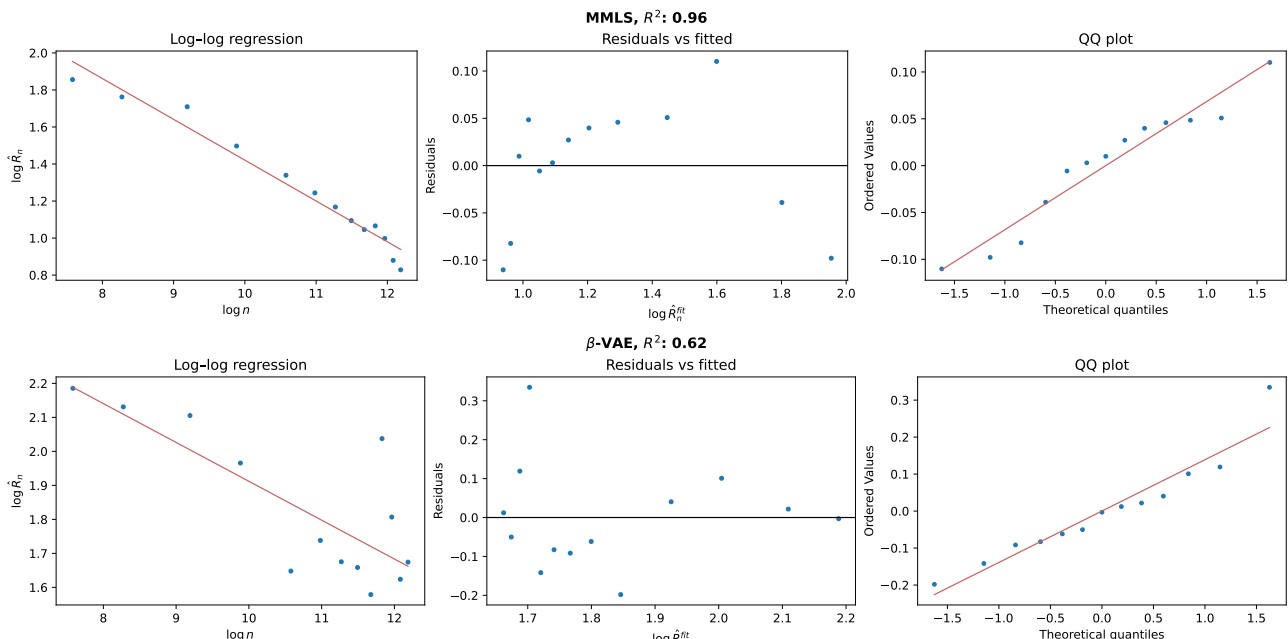

*Figure 27.* Regression evaluation for MMLS & $\beta$-VAE on dSprites (4D)

fitted manifold is not strongly distorted and the nearest projected point to $x_i$ is close to $\widehat{x}_i$.

We checked this approximation on the toy manifolds used in the fitting experiments by comparing $d_{\mathrm{pw}}$ with the full symmetric Hausdorff distance across sample sizes and repetitions. Figure 29 shows that the pointwise proxy follows the same trend as the full Hausdorff distance, with a mild overestimation in most regimes. Figure 30 shows the corresponding repetition-level comparison. The largest discrepancies occur in lower-sample regimes, where the fitted manifold is less stable.

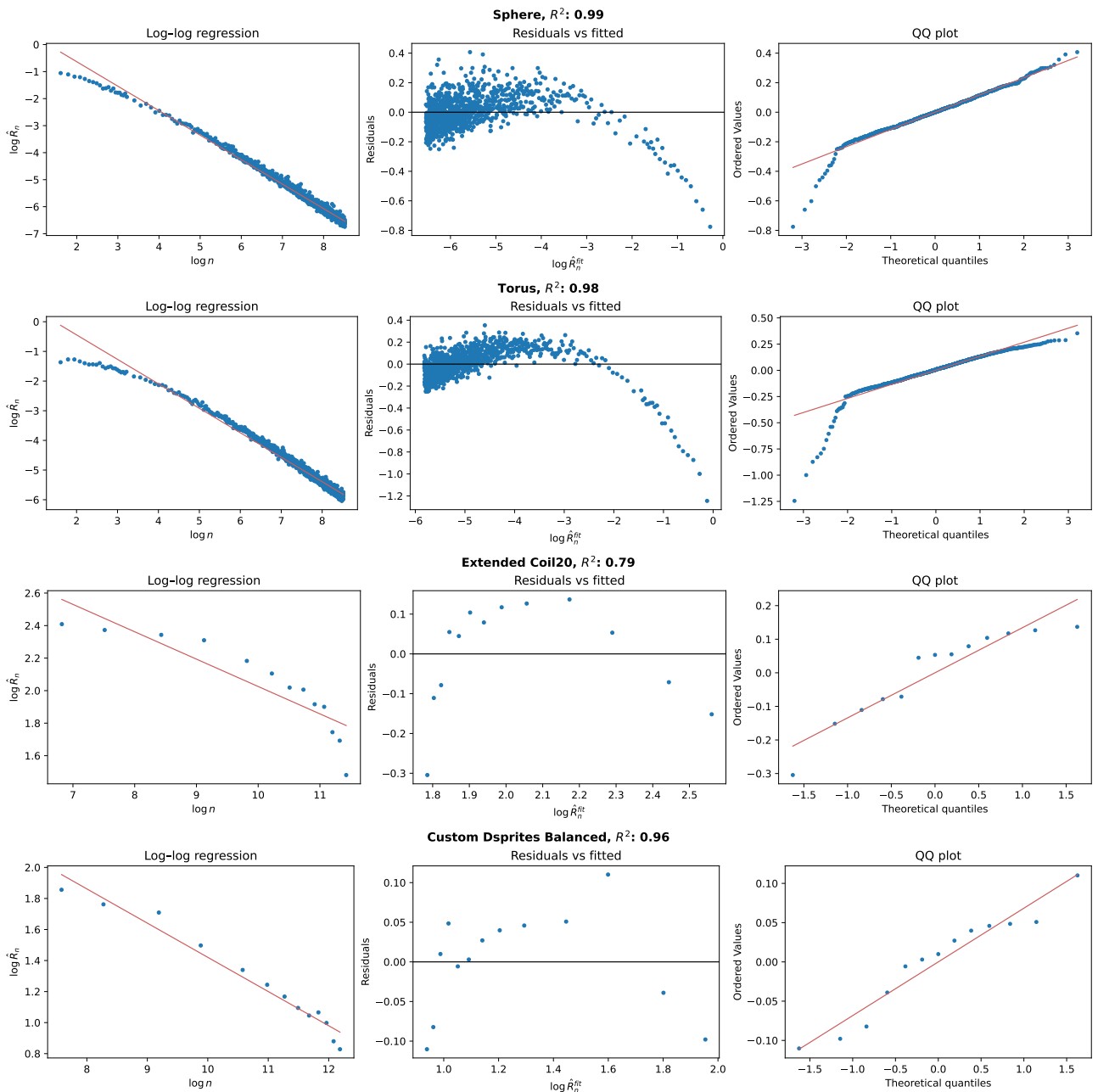

*Figure 28.* Regression evaluation for MMLS on from left to right Sphere, Torus, COIL-20 and dSprites.

## H. Per-Class Manifold Geometry on COIL-20

The bottom row of Figure 6 summarizes the COIL-20 results as the mean over the 20 object classes together with a $\pm 1\sigma$ band, in order to keep the main-paper figure legible. For completeness, Figure 31 reports the same six geometric measures for each object class *individually*.

The layout matches the main-paper figure: the $x$-axis traces the layers of the $\beta$-VAE from the input through the encoder to the latent $\mu$ and back through the decoder to the output, with the encoder and decoder halves shaded blue and orange respectively, and all measures normalized except volume. The per-class curves follow the same qualitative trends discussed in the main text, while exposing the inter-class spread that the $\pm 1\sigma$ band of Figure 6 aggregates into a single summary.

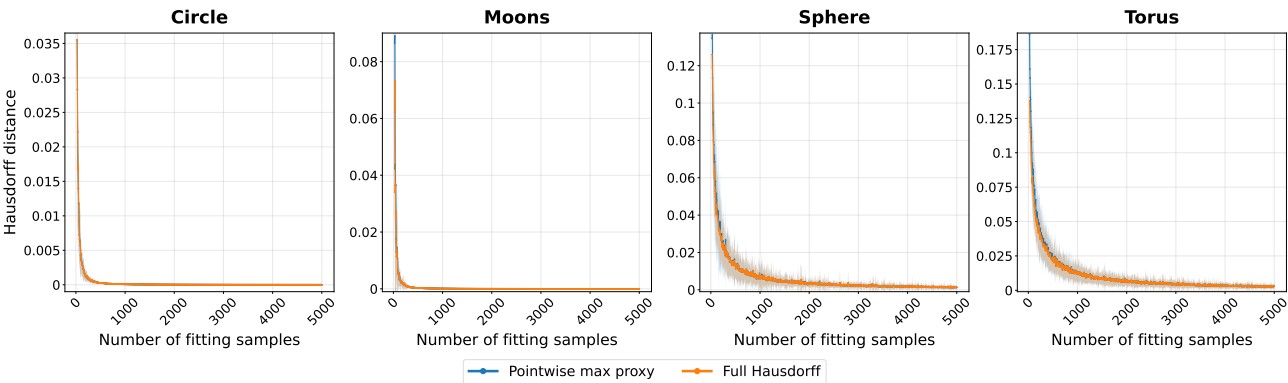

*Figure 29.* Comparison between the pointwise maximum proxy and the full symmetric Hausdorff distance on the toy manifold-fitting experiments. Curves show means over repetitions, with shaded regions corresponding to $\pm 2\sigma$.

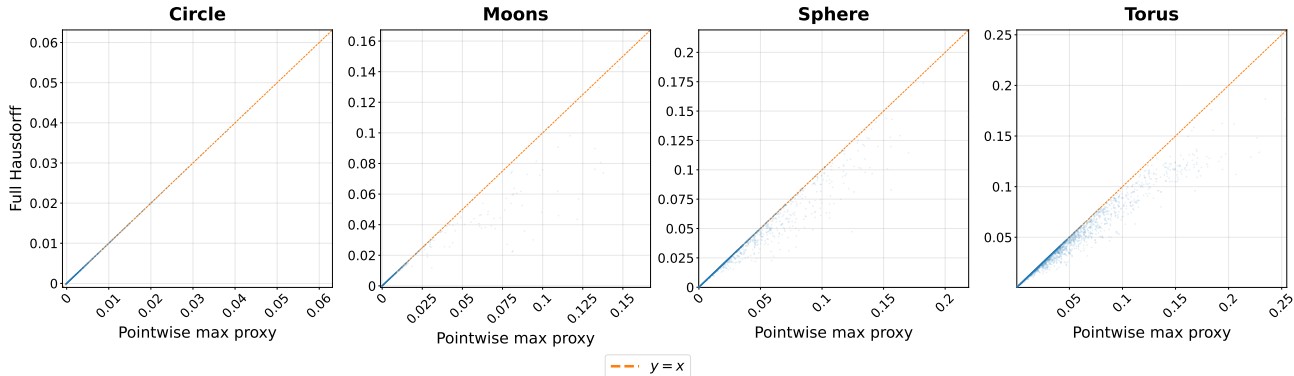

*Figure 30.* Repetition-level comparison between the pointwise maximum proxy and the full symmetric Hausdorff distance. The diagonal line indicates equality between the two quantities.

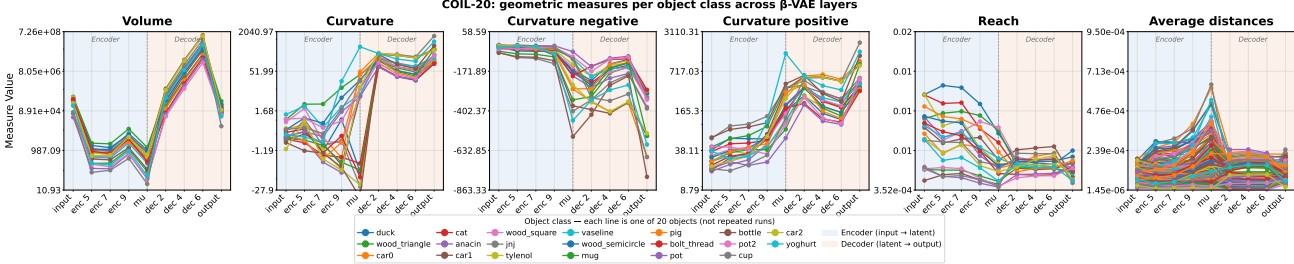

*Figure 31.* **Per-class COIL-20 manifold geometry across $\beta$-VAE layers.** Same six measures and layer layout as the bottom row of Figure 6, but each of the 20 curves is one COIL-20 object class (not repeated training runs). The $x$-axis runs input $\rightarrow$ encoder $\rightarrow$ latent $\mu \rightarrow$ decoder $\rightarrow$ output, with encoder and decoder halves shaded blue and orange. All measures are normalized except volume.

