# OpenReview forum: "The Data Manifold under the Microscope"
_ICML.cc/2026/Conference — ICML 2026 regular_

### Official Review · Reviewer_erP9 · 2026-03-10

**Soundness:** 3
**Presentation:** 3
**Significance:** 3
**Originality:** 3
**Overall Recommendation:** 4
**Confidence:** 3

**Summary:**

This work introduces a benchmarking framework to bridge the gap between deep learning theory and data manifold extraction in practice. The authors proposed a reproducible framework to process low dimensional  datasets with dense grid sampling. This provides a way to measure how the models fit real data manifolds. The authors also provided geometric estimators which are often difficult to estimate accurately in real world datasets.

**Compliance With Llm Reviewing Policy:**

Affirmed.

**Final Justification:**

The authors have resolved my concerns. I'd like to keep my positive rating.

**Key Questions For Authors:**

1. Can the authors justify why different transforms having an intense impact on the change of curvature directions?
2. Can the authors say more about the impact of a better geometric fit?
3. Can the authors comment a little on how to deal with the high dimensional case in the future works?

**Limitations:**

Yes

**Strengths And Weaknesses:**

Strengths:
1. This works proposes a new telescope to observe the gap between theoretical results on perfect manifolds and the practical messy datasets with unknown ground truth geometry.
2. The proposed high resolution grid sampling and high-order finite-difference estimators are shown to be more accurate than previous extrinsic estimators and easy to calculate.
3. This framework enables the authors to test previous theoretical results and track geometric evolution to see how a network processes data.

Weakness:
1. As the authors have pointed out in the discussion session, the current results only hold for dimensions 1-5. For high dimensional data it cannot be used while they are more general in practice.
2. The finite difference estimators require the sampling to be dense enough which makes it hard to scale efficiently and load with reasonable memory.
3. Besides scalability, some statements are made without clear enough justifications, e.g. the authors claim different transforms having an intense impact on the change of curvature direction, with no deep explanation of why specific transforms lead to noise in the curvature tensors.
4. It's still not clear if a better manifold fit would lead to a better downstream performance and how much this could help to improve.

---

> ### Author Rebuttal · Authors · 2026-03-30
>
> We thank Reviewer erP9 for the positive assessment and for the targeted questions. We address each concern below.
>
> **Can the authors justify why different transforms have an intense impact on the change of curvature directions?**
>
> We agree that this should be explained more clearly. The main reason is that different transforms do not act independently at second order. For combinations such as rotation and size, changing one transform can change the curvature induced by the other: for example, as size changes, the way pixel values evolve under rotation also changes, and the second-order rate of change can switch sign. This can produce alternating positive and negative sectional curvature contributions. If such terms dominate the scalar curvature, the signed scalar curvature plot becomes noisy due to cancellation, whereas taking absolute values gives a cleaner picture. Both views are informative, which is why we include both plots. We will revise the paper to explain this more explicitly and will try to add a concrete visual example, with a few transformed images and the corresponding effect on the curvature plots.
>
> **Can the authors say more about the impact of a better geometric fit?**
>
> This is an important question. In this paper we do not claim that better manifold fit automatically implies better downstream performance in general. Rather, our claim is that geometric fit is a useful quantity to measure, because it should matter in tasks where preserving reconstruction fidelity, local neighborhood structure, or class separation is important. For example, lower Hausdorff error should improve faithfulness of the learned manifold, and preserving reach or inter-class separation could plausibly support more stable downstream classification. However, the strength of this link is task- and model-dependent, and establishing it systematically is outside the scope of the current paper. One purpose of the benchmark is precisely to enable such future studies in a controlled setting.
>
> **Can the authors comment a little on how to deal with the high-dimensional case in future works?**
>
> We see the current framework mainly as a controlled low-dimensional benchmark, not as a direct solution to the high-dimensional case. There are several natural directions for future work. One is patchwise or chartwise application on manifolds with more complicated global structure but still low local dimension. Another is to use the benchmark to validate more general geometric estimators that work on unstructured point clouds, and then apply those estimators to higher-dimensional data where dense grids are infeasible. More broadly, the benchmark can serve as a calibration tool: methods intended for realistic high-dimensional settings can first be tested in a regime where the geometry is known.
>
> **The finite-difference estimators require the sampling to be dense enough, which makes it hard to scale efficiently and load with reasonable memory.**
>
> This is correct. Dense sampling is what makes the geometric quantities reliably computable, but it also imposes the main scalability limitation of the framework. The cost grows rapidly with intrinsic dimension, both in storage and in the number of points required for stable finite differences. We will make this trade-off more explicit in the discussion. Our view is that this is acceptable for a benchmark: its role is to provide controlled ground truth in the low-dimensional regime, against which more scalable methods can later be validated.
>
> **Some statements are made without clear enough justifications, e.g. the authors claim different transforms having an intense impact on the change of curvature direction, with no deep explanation.**
>
> We agree and refer to the discussion above. We will make this argument more explicit in the paper and, if space allows, include a concrete example showing how a pair of transforms can induce alternating curvature contributions and thus different behavior in the signed and absolute curvature plots.
>
> We appreciate these comments and believe they will help improve the paper. We will revise the discussion to better explain the geometric intuition behind the curvature observations, make the scope of the downstream-performance claims more precise, and more clearly position the framework as a controlled benchmark for developing and validating future geometric methods.

---

> > ### Author Rebuttal · Reviewer_erP9 · 2026-04-03
> >
> > The authors have solved my questions and I would like to keep my positive score.

---

> > > ### Author Response · Authors · 2026-04-03
> > >
> > > We would like to thank you once more for the constructive feedback and for taking the time to review our paper.

---

### Official Review · Reviewer_k1EA · 2026-03-12

**Soundness:** 3
**Presentation:** 3
**Significance:** 3
**Originality:** 3
**Overall Recommendation:** 5
**Confidence:** 4

**Summary:**

The authors aim to address a significant gap in the literature in the evaluation of generalization and approximation bounds in ML, more specifically, those that rely on the manifold hypothesis and derived quantities (curvature, reach or sampling density). According to the authors, current benchmarking options include either extremely simple manifolds, or real-world datasets where these quantities are unknown, making it difficult to really benchmark results.
To fill this gap, the authors propose a new benchmarking framework based on two existing datasets (dSprints amd COIL-20), which they enrich with additional transformation, along with finite-difference estimators of relevant manifold quantities. They then use their framework to evaluate theoretical bounds.
In section 3 of the paper, the authors introduce relevant quantities (e.g. definition of reach, Hausdorff distance, etc), before introducing known theoretical bounds on manifold estimation. In Section 4,  they present their estimation method. In Section 5, the authors showcase how the proposed framework can be applied for benchmarking theoretical bounds.

**Compliance With Llm Reviewing Policy:**

Affirmed.

**Final Justification:**

The authors have addressed my questions. I believe that this work will provide a useful benchmarking tool for comparing theoretical bounds. The authors have agreed to carefully discuss some of what I perceived to be limitations (e.g. convergence of the beta-VAE), etc.

**Key Questions For Authors:**

Figure 6:
a) The average distance plot for COIL-20 is absolutely unreadable — why are there so many curves out of the sudden?

b) Also, are the results averaged over different runs (e.g. initializations of the VAE)? How robust are these conclusions? Is it really judicious to plot the results using a line plot (eg. are all the points really ordered)? I’d personally split the plot between encoder and decoder, they are separate architectures, no? Especially since the encoder and decoder seem to have different behaviors (curve shapes)

c) I’m not sure what to make of the plots in this Figure: we see different quantities (reach, curvature, volume) being computed across layers. Is the point to say that the method proposed by the authors can be used to compute these metrics for the manifolds learned by every layer of a model? Is that a contribution? How does their method for computing, say, volume, compare to other method for evaluating manifold quantities?

Figure 3, 4,5:
Figure 5: we cannot see the Genovese upper bound in the sphere or torus,
I like the figures, but again, I’m not too sure what to make off of the results: “Across datasets and models, we find that the scaling implied by (Fefferman et al., 2018) is closer to the empirical error curves than the dimension-only rates of (Genovese et al., 2012).” Closer in what sense? Is the point that the shape of the empirical error matches more the shape of Fefferman?
The discussion on the VAE results is a little sparse: the authors attribute the lack of fit to some smoothing, but is it possible that the VAE is just not fitting correction? \beta-VAE has essentially a flat error as a function of n, indicating lack of consistency?
Im also confused about the authors’ discussion around sampling. They seem to suggest using two different types of sampling, but which one are they using in the experiments? Im also not sure what the role of sampling is, in this setting. Is it to estimate the quantities of interest (e.g reach) or to evaluate the bounds?

**Strengths And Weaknesses:**

**Strengths:**
Overall, the paper is clearly written and enjoyable to read. I find the topic interesting — while not being an expert on manifold estimation, I can definitely relate to the the problem of verifying the utility of a theoretical bound. The method seems detailed and compelling.


**Weaknesses:**
Despite my overall positive assessment of the paper, there are a few points that would benefit from further discussion, particularly in Section 5 where the authors showcase how to use their method. In particular, I find the figures hard to interpret and read (see questions below), and I think some of the contributions of the papers get lost in the presentation of the results. In particular, when showcasing in Figure 6 the different metrics for the two datasets, it is unclear if the computation of these quantities is a major contribution to the paper --- in which case they would need to be benchmarked (there is a bit of comparison done in Figure 2, but only on curvature). It is also unclear how the methe disintguishing between bounds (what is closeness)? The authors match bounds using a regression $log R ~ -A log(n) + C$ --- does closeness mean smaller $C$? Or a smaller distance (in what sense?) between the fitted curve and the empirical one? Should we report R2 for the regression?


Overall, this is an interesting paper that attempts to provide a clean benchmark and solution to compare theoretical bounds. However, some work is still left to do to show how, specifically, this framework can be used to make formal conclusions about these bounds.



*** Score increased from 4 to 5 after considering the authors' rebuttal.***

---

> ### Author Rebuttal · Authors · 2026-03-30
>
> We thank Reviewer k1EA for the careful reading and for concrete points that improve the paper’s clarity. We address the main questions below and will revise the presentation accordingly.
>
> **Figure 6a: The average distance plot for COIL-20 is absolutely unreadable — why are there so many curves?**
>
> This is a fair criticism. The many curves correspond to the 20 COIL-20 object classes, each defining a separate manifold tracked through the network layers; they are not different runs or initializations. We chose to show all classes to capture shared trends and class-to-class variability. However, we agree that the current presentation is too crowded for the main paper. We will simplify the figure, improve the legend/caption, and move the full class-wise plot to the appendix.
>
> **Figure 6b: Are the results averaged over runs? Are line plots appropriate? Should encoder/decoder be split?**
>
> These results are not averaged over different VAE initializations; they come from one trained $\beta$-VAE per dataset. The x-axis is ordered by network depth, so the line plot shows how the geometry evolves layer by layer. We agree, however, that the encoder and decoder should be separated more clearly, since they play different roles and show different trends. We will revise the plot accordingly and make explicit in the text that the curves represent manifold classes across layers, not repeated runs.
>
> **Figure 6c: Is the point that your method can compute volume/reach/curvature for every layer? Is that a contribution? How does it compare to other methods?**
>
> Yes, one contribution of the framework is precisely that, once a benchmark manifold with dense known sampling is available, one can track geometric quantities such as volume, curvature, reach, and class separation through intermediate network representations. The main contribution is therefore the benchmarking framework that makes such measurements possible in a controlled setting; the $\beta$-VAE analysis is an illustrative use case. On the estimator side, our curvature estimator is already compared on analytic manifolds in Figure 2. More broadly, the novelty is not a universal new estimator outperforming all alternatives, but a controlled setup in which such estimators can be validated and used reliably.
>
> **Figures 3–5: Closer in what sense? Should $R^2$ be reported?**
>
> We agree this should be stated much more clearly. By closer we mean closer in scaling exponent: we fit a power law $C n^{-A}$ to the empirical Hausdorff error curve and compare the fitted exponent $A$ to the exponents predicted by the different theories. Thus the comparison is about agreement of the log-log slope, not about absolute curve overlap. We will make this explicit. The full regression analysis, including the $R^2$ values, is already in the appendix; we also plan to add a small table to the main paper with the $R^2$ values for the three experiments, so fit quality is visible without consulting the appendix.
>
> **Could the $\beta$-VAE simply not be fitting correctly?**
>
> Yes, that is a real possibility, and our current discussion should be more careful on this point. The near-flat error curve suggests that, in the tested regime, the $\beta$-VAE does not exhibit a clear asymptotic improvement with sample size. This could reflect the KL regularization, but also optimization limits or underfitting of the chosen architecture/training setup. We will tone down the interpretation accordingly. In our view, this is still a useful outcome of the benchmark: it reveals when a model does not follow the theoretically expected scaling.
>
> **I’m confused about sampling: which sampling is used, and what role does it play?**
>
> There are two uses of sampling, and we will separate them more clearly in the paper. First, the geometric quantities of the benchmark datasets are computed once on the densest available grid using finite differences. Second, for the manifold-fitting experiments, we draw training subsets of size $n$ intended to approximate uniform sampling on the manifold. For analytic manifolds this is done directly from the closed-form parametrization; for the image manifolds, we subsample the dense grid with weights proportional to the local volume element. Thus, sampling is used to evaluate the fitting bounds, not to define the benchmark geometry itself.
>
> **We cannot see the Genovese upper bound in the sphere or torus.**
>
> For the sphere and torus, the intrinsic dimension is $d=2$, and the two reference exponents coincide: $\frac{2}{d+2}=\frac{1}{d}=\frac{1}{2}$. So in those panels the Genovese and Fefferman reference curves overlap visually. We will adjust the plotting/commentary so this is not confusing.
>
> We appreciate these comments and believe they will help improve the paper. We will revise the text and figures to better highlight the benchmark contribution, clarify the exponent comparison, and make the role of the $\beta$-VAE example and the different sampling procedures more explicit.

---

> > ### Author Rebuttal · Reviewer_k1EA · 2026-04-02
> >
> > Thank you for addressing my comments, and I look forward to reading the revised version of the manuscript.
> > I think this addresses my comments and I'll increase my score.
> >
> > The only minor reservation I have is from the VAE side: "The near-flat error curve suggests that, in the tested regime, the $\beta$-VAE does not exhibit a clear asymptotic improvement with sample size. This could reflect the KL regularization, but also optimization limits or underfitting of the chosen architecture/training setup. We will tone down the interpretation accordingly. In our view, this is still a useful outcome of the benchmark: it reveals when a model does not follow the theoretically expected scaling": I am not quite sure how to interpret the deviation from the expected scaling though. In my understanding, most error bounds are usually of the form stochastic error + optimization error -- and if the expected scaling doesn't kick in (or even in this case, we never see any sign of consistency), this could be more a sign of optimization error? In this case, would the utility be in diagnosing this effect? This tells us more about the optimization side of things than the limits of the method though...

---

> > > ### Author Response · Authors · 2026-04-02
> > >
> > > We thank you again for this helpful observation. Indeed, in the tested regime, the near-flat $\beta$-VAE curve may indicate optimization or regularization effects rather than the statistical limit of the method itself. We will update the text to make this clear and to frame the result as a diagnostic use of the benchmark, as you also suggest: in this setup, a clear asymptotic scaling regime is not yet visible, so comparisons to manifold-fitting theory should be interpreted with appropriate care.

---

### Official Review · Reviewer_vJLs · 2026-03-12

**Soundness:** 3
**Presentation:** 3
**Significance:** 3
**Originality:** 3
**Overall Recommendation:** 4
**Confidence:** 2

**Summary:**

This paper proposes a controlled benchmarking framework for studying the geometry of data manifolds in deep learning. The authors construct densely sampled, low-dimensional manifolds, both analytic (e.g., spheres, tori) and adapted image datasets (extended dSprites and COIL-20), with known intrinsic factors and grid structure. This enables accurate finite-difference estimation of geometric quantities such as volume, scalar curvature, and reach. The framework is used for two main applications, empirical evaluation of theoretical manifold-fitting bounds (Genovese et al., Fefferman et al.) via Hausdorff error scaling and analysis of how geometry evolves across layers of β-VAEs, tracking changes in curvature, reach, and inter-class separation. The authors attempt to address the empirical grounding of manifold-based theory  where geometry can be computed reliably and used to stress-test existing bounds.

**Compliance With Llm Reviewing Policy:**

Affirmed.

**Final Justification:**

Overall, the authors discuss a pressing challenge in bridging geometric theory and empirical validation, and the authors attempt to address an important concept through a controlled benchmarking framework.
The rebuttal clarified the intended benchmark-focused contribution and explicitly acknowledged the main limitations, which addresses my primary concerns. While these limitations remain, they are now appropriately scoped, and this clarification improves my overall evaluation of the paper.

**Key Questions For Authors:**

1. Do you expect the curvature/reach trends observed in β-VAE to generalize to diffusion models or other generative architectures?
2. How sensitive are curvature and reach estimates to grid resolution in the image-based datasets?
3. Have you tested how robust the geometric measures are to small perturbations or noise in the grid?

**Limitations:**

yes

**Strengths And Weaknesses:**

Strengths:


1. The gap between geometric theory and practice is well articulated. Many manifold-based guarantees rely on parameters that cannot be measured in real data. Providing a controlled setting where these quantities are accessible is useful.

2.  The scope is necessarily limited to low-dimensional manifolds, but within that scope, the framework is convincing and well validated. For researchers working at the intersection of geometry and representation learning, this could be a genuinely useful tool.


3. Even beyond the specific experiments, the framework itself could be valuable for validating curvature/reach estimators or stress-testing geometric assumptions in other models.

Weaknesses:


1. The framework only works for very low intrinsic dimensions (roughly up to 4–5). While this is acknowledged, it limits how far we can extrapolate the findings to realistic high-dimensional generative models.

2. The datasets are still highly controlled and mostly axis-aligned in their latent factors. This makes the geometry relatively “clean.” It’s not clear how well conclusions carry over to more entangled or naturally occurring manifolds.

3. While conceptually strong, the work is more of an infrastructure/benchmark contribution. Only MMLS and β-VAE are evaluated. It would strengthen the paper to include at least one modern generative model (e.g., diffusion) to test whether geometric trends persist.

---

> ### Author Rebuttal · Authors · 2026-03-30
>
> We thank reviewer vJLs for the thoughtful feedback. We agree that the paper should be read primarily as a benchmark/framework contribution: its goal is to provide a controlled setting in which geometric quantities are measurable and methods can be stress-tested, rather than to claim immediate transfer of the findings to arbitrary real-world datasets.
>
> **The framework only works for very low intrinsic dimensions (roughly up to 4–5). While this is acknowledged, it limits how far we can extrapolate the findings.**
>
> We agree this is the main limitation and we will make it even more explicit. The conclusions should indeed not be overgeneralized. Our intent is rather to provide a high-quality benchmark where geometric ground truth is accessible and failure modes of methods can be identified before moving to harder settings. In that sense, the framework is closer to a controlled test suite than to a fully general-purpose methodology.
> This low-dimensional regime is still meaningful: dSprites ($d=4$) contains nontrivial visual transformations, and COIL-20 ($d=3$) captures real object appearance variation. More broadly, many theoretically motivated results in manifold learning have not been empirically examined even in such controlled settings. We believe there is value in first validating them where the relevant geometric quantities can actually be computed.
>
> **The datasets are still highly controlled and mostly axis-aligned in their latent factors. It’s not clear how well conclusions carry over to more entangled or naturally occurring manifolds.**
>
> This is a fair point, and we will clarify it further. The clean, grid-structured latent parametrization is a deliberate design choice that allows us to compute geometric quantities reliably via finite differences; it is not meant to model the latent organization of arbitrary real datasets. At the same time, the framework is not restricted to globally simple topology: for low-dimensional manifolds with more complicated global structure, it can in principle be applied patchwise on an atlas, with the geometry computed separately on each chart. This still places the framework in the regime of roughly 4-5-dimensional manifolds, but even there one can already realize mathematically nontrivial phenomena and test how methods behave in their presence, rather than only on the simplest canonical shapes.
>
> **It would strengthen the paper to include at least one modern generative model (e.g., diffusion) to test whether geometric trends persist.**
>
> We agree this would be a valuable extension. We chose the $\beta$-VAE because it provides a particularly natural first case study: the decoder gives an explicit low-dimensional representation map, making it easier to analyze learned geometry in a controlled way. Extending the framework to diffusion models is an important next step, but we felt that including it here would substantially broaden the scope beyond a single paper. We will clarify that the $\beta$-VAE part is intended as one worked example of how the benchmark can be used.
>
> **How sensitive are curvature and reach estimates to grid resolution in the image-based datasets?**
>
> In general, for smooth manifolds and central finite differences, the discretization error scales as $O(h^2)$, where $h$ is the grid spacing. This is also consistent with our validation on analytic manifolds with known geometry, where increasing the grid density improves the estimates. For the image-based datasets, however, we have not yet performed a separate systematic ablation focused only on grid resolution, so we do not want to overclaim robustness there. We are planning to run such an ablation for the final version of the paper.
>
> **Have you tested how robust the geometric measures are to small perturbations or noise in the grid?**
>
> We did not test this systematically. Our expectation is that adding noise to the manifold samples would tend to increase estimated volume and curvature and decrease reach, since the manifold becomes locally rougher and effectively thicker. This is, however, outside the main scope of the paper: our goal is to provide benchmark datasets with fixed, known geometry, against which one can then test the robustness of other methods. In such a use case, noise would be applied to the benchmark data or to the method under test, while the reference geometric properties of the underlying dataset manifold would still be computed once from the clean benchmark construction.
>
> We will revise the paper to better emphasize this benchmark-oriented scope, make the limitations more explicit, and more clearly separate what is a framework contribution from what is an illustrative empirical insight.

---

> > ### Author Rebuttal · Reviewer_vJLs · 2026-04-03
> >
> > My concerns have been sufficiently addressed through clarification of scope and limitations, so I am raising my score accordingly.

---

> > > ### Author Response · Authors · 2026-04-04
> > >
> > > We thank the reviewer for their time, for highlighting interesting directions for future work, and for updating their score.

---

### Official Review · Reviewer_k76t · 2026-03-14

**Soundness:** 3
**Presentation:** 3
**Significance:** 3
**Originality:** 3
**Overall Recommendation:** 5
**Confidence:** 4

**Summary:**

This paper introduces a benchmark framework for studying data manifolds in a controlled yet nontrivial setting. The authors construct densely sampled low-dimensional image manifolds (including adapted versions of dSprites and COIL-20), estimate geometric quantities such as volume, curvature, and reach using the known grid structure, and use the resulting setup to evaluate manifold fitting theory and analyze learned representations in β-VAEs. The main contribution is an experimental framework that makes geometric properties measurable enough to empirically test theoretical predictions and inspect how representation geometry changes across network layers.

**Compliance With Llm Reviewing Policy:**

Affirmed.

**Final Justification:**

The authors have solved my questions and I would like to keep my positive score.

**Key Questions For Authors:**

1.The empirical comparison to theory relies on fitting constants from the observed curves. Can the authors clarify how strongly this should be interpreted as evidence for the practical usefulness of the original bounds?

2.How robust are the β-VAE findings across architecture, latent dimension, β, and random seed?

3.The paper uses directed Hausdorff distance as a computable proxy. How often does this differ qualitatively from the full Hausdorff distance in practice?

4.What are the main bottlenecks to extending the framework to higher intrinsic dimensions or more complex manifold structure?

**Limitations:**

The paper does acknowledge the main limitations, especially the restriction to low-dimensional and relatively simple manifolds. I think this discussion is adequate, though the paper could be a bit more explicit that findings from this benchmark may not directly transfer to more realistic representation-learning settings. I do not see major unaddressed societal concerns.

**Strengths And Weaknesses:**

Soundness
The paper is technically careful overall. The benchmark construction is well motivated, the geometric estimators are clearly defined, and the appendix provides useful derivations and implementation details. The experiments are reasonably systematic across both synthetic and image-based manifolds.
That said, the comparison to theory is somewhat limited because constants are fit post hoc from empirical curves, so the results mainly validate relative scaling behavior rather than the practical predictive power of the original bounds. Some conclusions, especially in the β-VAE part, are also somewhat noisy and should be interpreted with care.

Presentation
The paper is generally well written and easy to follow. The structure is sensible, and the figures are effective in illustrating both the benchmark and the applications.
My main issue is that some important experimental details and caveats are pushed to the appendix. The main text could also more clearly separate what is a benchmark contribution versus what is a new empirical or theoretical insight.

Significance
I think the paper addresses an important problem: connecting manifold-based theory with empirical deep learning practice. A controlled benchmark of this kind could be useful for future work in manifold learning, representation learning, and geometric ML.
The main limitation is scope: the framework currently applies to relatively low-dimensional and simple manifolds, so the conclusions should not be overgeneralized to more realistic large-scale settings.

Originality
The work is original mainly as a benchmark/framework contribution. It does not propose a fundamentally new learning algorithm or theorem, but it combines dataset construction, geometric estimation, and representation analysis in a way that I found novel and useful. The adapted image-manifold benchmarks are a meaningful part of the contribution.

---

> ### Author Rebuttal · Authors · 2026-03-30
>
> We thank reviewer k76t for the constructive and encouraging review. We address the main questions below and will revise the paper accordingly.
>
> **Q1: How should the comparison to theory be interpreted, given that constants are fit post hoc?**
>
>  Our intent is not to claim practical prediction from the original bounds via their constants. The main evidence is about scaling behavior: whether the observed error curves are consistent with the rates suggested by the theory. This is the aspect that remains meaningful in our setting. For the ambient dimensions relevant to image manifolds, the constants in existing bounds are extremely loose and not practically informative, so we estimate them empirically. We will make this limitation more explicit in the main text and state clearly that our results support rate-level agreement rather than practical constant-level predictivity.
>
>
> **Q2: How robust are the $\beta$-VAE findings across architecture, latent dimension, $\beta$, and seed?**
>
>  We did not run a full ablation, so we do not want to overstate robustness here. We used the standard $\beta$-VAE setup from the original public implementation, adapting only minor training hyperparameters such as batch size and learning rate for our hardware. In a small number of additional qualitative runs, the trends looked similar, but this is not sufficient to claim robustness across architectures or hyperparameters. We will clarify that the $\beta$-VAE section should be read as a worked example / case study of how the benchmark can be used, not as a broad claim about $\beta$-VAEs in general.
>
> **Q3: How different is the proxy from the full Hausdorff distance in practice?**
>
>  We had not evaluated this comparison before submission, but we have now run it on the toy manifolds where the full Hausdorff distance is computationally feasible. The proxy and full Hausdorff curves overlap very closely overall, with the largest deviations appearing only in the low-sample regime for the torus. We plan to include the overlap and error plots in the appendix. As representative numbers, the mean absolute relative error is $0.00550$ on the sphere and $0.0114$ on the torus.
>
> **Q4: What are the main bottlenecks for extending the framework to higher intrinsic dimension or more complex manifolds?**
>
>  The main bottleneck is the need for dense grid sampling to compute finite differences reliably. This leads to exponential growth in the number of required samples with intrinsic dimension. A natural next step would be to replace the grid-based estimators by methods that operate on unstructured point clouds, including recent geometric estimators from the manifold estimation literature. One motivation for our benchmark is precisely to provide a controlled testbed for validating such methods before applying them to higher-dimensional data. For more complex topology, the framework can in principle still be applied patchwise on an atlas, provided each patch can be sampled densely enough.
>
> We also agree with the presentation-related comments. In the revision we will move a few key experimental details/caveats from the appendix to the main text, better separate the benchmark contribution from the empirical insights, and add a stronger warning that conclusions from this benchmark should not be overgeneralized to large-scale high-dimensional representation-learning settings.

---

> > ### Author Rebuttal · Reviewer_k76t · 2026-04-01
> >
> > My problem has been basically solved and I maintain my previous score

---

> > > ### Author Response · Authors · 2026-04-02
> > >
> > > We thank the reviewer again for their time and feedback. We are glad that the main concerns have been resolved, and we will incorporate the clarifications in the revision.

---

### Decision · Program_Chairs · 2026-04-30

**Decision:**

Accept (regular)

**Comment:**

The authors provide a benchmark for the study of data geometry in ML by extending dSprites and COIL-2 with denser sampling. This provides an accurate estimation of curvature, reach, and volume, and one can assess bounds from previous theory-oriented results, as well as variational autoencoder geometry.

All the reviewers found the results very interesting, novel, and original. They are well motivated and well presented, and relevant and of great interest to the ICML community, as a comprehensive benchmark on this topic has been missing. I also agree with them, and I suggest acceptance.